# Repression of RNA polymerase by the archaeo-viral regulator ORF145/RIP

Carol Sheppard[1], Fabian Blombach[1], Adam Belsom[2], Sarah Schulz[3], Tina Daviter[1], Katherine Smollett[1], Emilie Mahieu[1], Susanne Erdmann[4], Philip Tinnefeld[3], Roger Garrett[4], Dina Grohmann[3,†], Juri Rappsilber[2,5] & Finn Werner[1]

Little is known about how archaeal viruses perturb the transcription machinery of their hosts. Here we provide the first example of an archaeo-viral transcription factor that directly targets the host RNA polymerase (RNAP) and efficiently represses its activity. ORF145 from the temperate *Acidianus* two-tailed virus (ATV) forms a high-affinity complex with RNAP by binding inside the DNA-binding channel where it locks the flexible RNAP clamp in one position. This counteracts the formation of transcription pre-initiation complexes *in vitro* and represses abortive and productive transcription initiation, as well as elongation. Both host and viral promoters are subjected to ORF145 repression. Thus, ORF145 has the properties of a global transcription repressor and its overexpression is toxic for *Sulfolobus*. On the basis of its properties, we have re-named ORF145 RNAP Inhibitory Protein (RIP).

[1] Division of Biosciences, Institute of Structural and Molecular Biology, University College London, London WC1E 6BT, UK. [2] Wellcome Trust Centre for Cell Biology, University of Edinburgh, Edinburgh EH9 3BF, UK. [3] Physikalische und Theoretische Chemie—NanoBioSciences, Technische Universität Braunschweig, Hans-Sommer-Strasse 10, Braunschweig 38106, Germany. [4] Department of Biology, University of Copenhagen, Copenhagen DK-2200, Denmark. [5] Department of Bioanalytics, Institute of Biotechnology, Technische Universität Berlin, Berlin 13355, Germany. † Present address: Department of Microbiology and Archaea Centre, Laboratory of Single-Molecule Biochemistry, University of Regensburg, Regensburg 93053, Germany. Correspondence and requests for materials should be addressed to F.W. (email: f.werner@ucl.ac.uk).

Viruses and bacteriophages have been valuable model systems for studying gene regulation since the advent of molecular biology[1,2], and they play an important role in evolution by facilitating lateral gene transfer[3]. Cellular organisms belonging to the eukaryotes, archaea or bacteria are accompanied by their cognate viruses (termed phages in bacteria)[4]. Viruses and phages have coevolved with their hosts, are restricted to their cognate domain of life and display only limited sequence conservation. While phages have been studied since the 1950s, and eukaryotic pathogenic viruses are the focus of much biomedical research, the viruses accompanying the archaea have not been comprehensively studied yet. Archaeal viruses exhibit unique and diverse morphologies[4,5]. One example is the Sulfolobales-infecting Acidianus two-tailed virus (ATV), which has the ability to undergo morphological changes outside the host cell, forming long bipolar tails that aid in host cell attachment in a low-cell-density environment[6]. ATV is a temperate virus with lysogenic and lytic life stages depending on environmental cues, chiefly suboptimal growth temperatures[6]. Despite the growing numbers of morphologic and genomic studies of archaeal viruses, a detailed investigation of their molecular processes such as replication and transcription as well as host–virus relationships remain limited[7]. The ongoing battle between archaea and their viruses is reflected in the high abundance of CRIPSR-Cas adaptive immune systems in archaea[8]. In a systematic survey of CRISPR-Cas systems in Sulfolobales, spacers directed against ATV were identified in all genomes[9].

In all three cellular domains of life, transcription is carried out by highly conserved multisubunit RNA polymerases (RNAPs)[10]. The archaeal RNAP and eukaryotic RNAPII require the basal transcription factors TBP (TATA-binding protein) and TFIIB (transcription factor IIB in RNAPII and TFB in archaea) to preassemble on the TATA and BRE (B-recognition element) motifs of the promoter prior to RNAP recruitment. A third factor, TFE (TFIIE), stimulates transcription initiation by facilitating DNA melting[11–13]. During this process, the flexible RNAP clamp opens to allow the loading of the template strand into the active site. In contrast, during elongation the clamp is closed to prevent premature dissociation of the elongation complex, while transcription termination is likely to require a transient opening of the clamp. Interactions between general transcription factors and RNAP modulate the position of the clamp; TFE binding to RNAP favours opening of the clamp[14].

While the minimal complement of basal transcription factors in archaea mirrors the RNAPII system, gene-specific transcription factors are diverse and employ regulatory mechanisms prototypical for bacterial and eukaryotic factors. Known archaeal transcription repressors act via promoter occlusion as in bacteria[15,16], while transcription activators act via augmented recruitment of basal initiation factors TBP and TFB[17,18], reminiscent of some eukaryotic transcription activators. Viral and phage transcription factors often appropriate their host's gene expression machinery for their own purposes[1,19]. A limited number of archaeo-viral transcription factors have been identified and functionally characterized including SvtR, AvtR and F55, which are all DNA-binding and gene-specific[20–23]. In contrast, several phages encode transcription factors that directly target RNAP and have the power to influence transcription on a genome-wide level[24]. For example, the phage T7-encoded gp2 protein is a global repressor. Gp2 is expressed during early T7 infection of Escherichia coli, where it binds to and represses the host-encoded RNAP, thereby efficiently shutting down the entire host transcriptome[25,26]. Other early expressed genes include the T7 RNAP, which is inert against gp2 action and which directs transcription of the middle and late-phage genes that are under the control of T7 promoters[27]. Thus, the phage redirects the remaining host resources for its own purposes. Apart from chromatin proteins such as histones, global regulators of transcription have not been found in archaea or their viruses, and none of the sequenced archaeo-viral genomes encodes (recognizable) RNAPs.

This study focuses on a novel regulator encoded by ATV. ORF145 encodes an abundant 145 amino-acid (16.8 kDa) protein that was identified as a virion protein and putative RNAP interactor in a screen of ATV-encoded gene products[6]. Here we undertake a multidisciplinary structural and functional characterization of ORF145 to unravel its role in transcription regulation. Our results demonstrate that the protein (i) directly binds to RNAP with high affinity, which (ii) prevents the formation of transcription pre-initiation complexes (PICs), (iii) represses abortive and productive initiation and (iv) represses transcription elongation. We propose a mechanism by which ORF145 is wedged into the DNA-binding channel of RNAP and locks the otherwise flexible clamp into one fixed position. In agreement with its in vitro characteristics, the homologous expression of ORF145 in Sulfolobus is highly toxic. On the basis of its properties we name ORF145 RNAP inhibitory protein, RIP for short and we refer to the protein as ORF145/RIP throughout the manuscript.

## Results

**Formation of a high-affinity complex with Sso RNAP.** The ATV ORF145/RIP gene product was initially identified as RNAP-binding protein in an interaction screen of unannotated ATV proteins. In order to validate and characterize this interaction, we produced recombinant ORF145/RIP and tested its interaction with purified Sulfolobus solfataricus (Sso) RNAP using size exclusion chromatography (SEC). ORF145/RIP elutes as a single peak corresponding to a molecular weight of ∼17 kDa demonstrating that ORF145/RIP is monomeric (Fig. 1a, red trace). When ORF145/RIP was pre-incubated with RNAP and subjected to SEC, an additional peak appeared in the high molecular weight range corresponding to ∼400 kDa in good agreement with the size of the Sso RNAP (406.8 kDa; Fig. 1a, blue trace). Immuno-detection confirmed the presence of both Sso RNAP and ORF145/RIP in fractions corresponding to the ∼400 kDa peak (Fig. 1a and Supplementary Fig. 1A), indicating the formation of ORF145/RIP–RNAP complexes. In order to quantify the binding affinity, we produced a [32]P-labelled variant of ORF145/RIP and tested its interaction with RNAP in electrophoretic mobility shift assays (EMSA). [32]P-ORF145/RIP is resolved as a single distinct high-mobility band, while incubation with increasing amounts of RNAP led to the upshift of the signal to a new low-mobility band in a dose–response dependent manner (Fig. 1b). Quantification of this signal presented in a saturating binding curve yielded a dissociation constant of $12.6 \pm 1.9$ nM indicative of a high-affinity interaction (Fig. 1b).

To test whether electrostatic interactions facilitated the ORF145/RIP binding to RNAP, we performed EMSAs with increasing concentrations of salt (Supplementary Fig. 1B). The ORF145/RIP–RNAP complexes were resistant to high ionic (2 M NaCl) strengths, indicating that the ORF145/RIP–RNAP interaction is not exclusively based on electrostatic interactions.

**ORF145/RIP interacts with the RNAP clamp.** In order to identify which of the 13 subunits of the Sso RNAP contribute to the ORF145/RIP binding, we carried out far-western blotting experiments. The subunits of purified Sso RNAP, as well as a cell-free Sso extract were resolved by SDS–polyacrylamide gel electrophoresis (SDS–PAGE), transferred to a membrane, incubated with ORF145/RIP, washed and subsequently probed

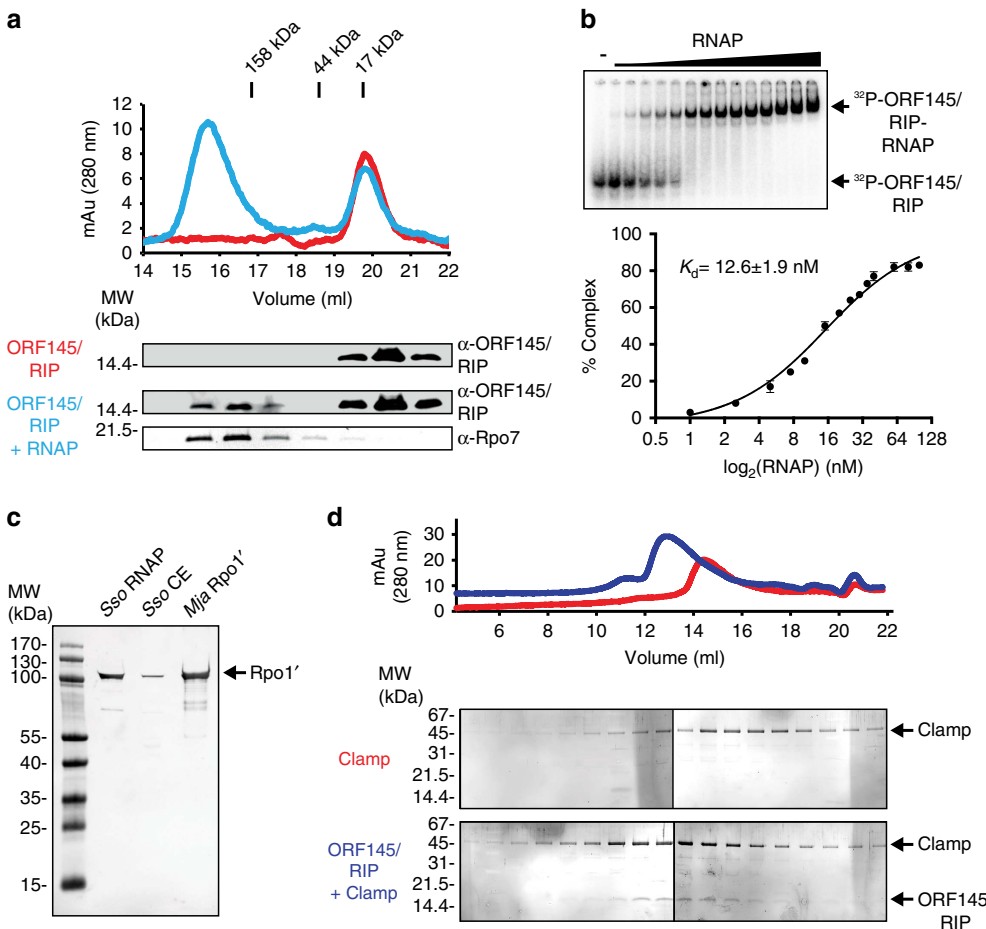

**Figure 1 | ORF145/RIP binds to the Sso RNAP via the Rpo1' clamp. (a)** SEC of ORF145/RIP and ORF145/RIP–RNAP complexes. Ultraviolet elution profiles of ORF145/RIP (red trace) and ORF145/RIP pre-incubated with Sso RNAP (blue trace). Western blot analysis of SEC peak fractions detecting 'free' ORF145/RIP (elution volume 19–21 ml), and co-elution of ORF145/RIP with Sso RNAP (elution volumes 15–17 ml). **(b)** EMSA of the ORF145/RIP–RNAP complex using radiolabelled ORF145/RIP. The gel bands were quantified and then plotted with the tight binding equation (see Methods for further details) from which a $K_d$ of $12.6 \pm 1.9$ nM was calculated. **(c)** A far-western blot was conducted by immobilizing purified RNAP and cell extract samples, probing first with recombinant ORF145/RIP protein followed by anti-ORF145/RIP polyclonal serum and finally anti-rabbit secondary antibody. ORF145/RIP interacts with the Rpo1' subunit from the purified 13-subunit Sso RNAP, Sso cell extract, as well as purified recombinant Mja Rpo1' subunit. **(d)** SEC analysis of the recombinant RNAP clamp domain (red trace) and the ORF145/RIP–clamp complex (blue trace). Error bars represent standard deviation from three technical repeats.

with polyclonal antibodies raised against ORF145/RIP (Fig. 1c). The far-western blot shows a single band with a mobility of ∼100 kDa in lanes containing purified RNAP or cell extract, which corresponds to the largest RNAP subunit Rpo1' (Fig. 1c). We tested the species specificity of the Rpo1'–ORF145/RIP interaction using recombinant Rpo1' protein from the euryarchaeon *Methanocaldococcus jannaschii* (Mja) in far-western blots. Both Sso and Mja RNAP Rpo1' yield a comparable signal, which suggests that the ORF145/RIP-binding site resides in a highly conserved domain of Rpo1' (Fig. 1c). A negative control experiment excluding incubation with ORF145/RIP ruled out nonspecific binding of the antibody (Supplementary Fig. 1C).

Rpo1' is the largest RNAP subunit and includes among other motifs the universally conserved triple-aspartate loop of the active site, the RNAP clamp and part of the DNA-binding channel. The clamp is a regulatory 'hotspot' and binding site for the DNA template and for the basal factors TFE and Spt4/5 (refs 12,14). In order to test whether ORF145/RIP interacts with the clamp, we generated a synthetic recombinant RNAP clamp composed of fragments of the Rpo1', Rpo1'' and Rpo2 subunits expressed as a single polypeptide chain[11,28]. The clamp elutes as a broad peak from a SEC column centred at an elution volume of 14.3 ml

(Fig. 1d, red trace). Combining ORF145/RIP and the clamp prior to SEC leads to a shift of the peak to 12.8 ml (Fig. 1d, blue trace). Sypro-Ruby-stained SDS–PAGE analysis shows that ORF145/RIP is co-eluting symmetrically with the RNAP clamp, suggesting that they form a stable complex at room temperature (Fig. 1d).

**ORF145/RIP binds in the DNA-binding channel.** To further map the interaction of ORF145/RIP with RNAP, we used a combination of chemical crosslinking and mass spectrometry. This method has been successfully applied to identify the binding site of TFIIF on yeast RNAPII (ref 29,30). Recombinant ORF145/RIP and purified RNAP were incubated to allow complex formation and chemically crosslinked using the lysine-specific crosslinker bis(sulfosuccinimidyl) suberate (BS3) that preferentially reacts with the side chains of lysine residues but also those of serine, threonine and tyrosine, albeit to a lesser extent. The crosslinked material was subjected to SDS–PAGE and yielded a single band with low electrophoretic mobility compared with the (uncrosslinked) individual polypeptides of ORF145/RIP and RNAP (Fig. 2a). The crosslinked complex was isolated from the gel and treated with trypsin, and the resulting peptide fragments

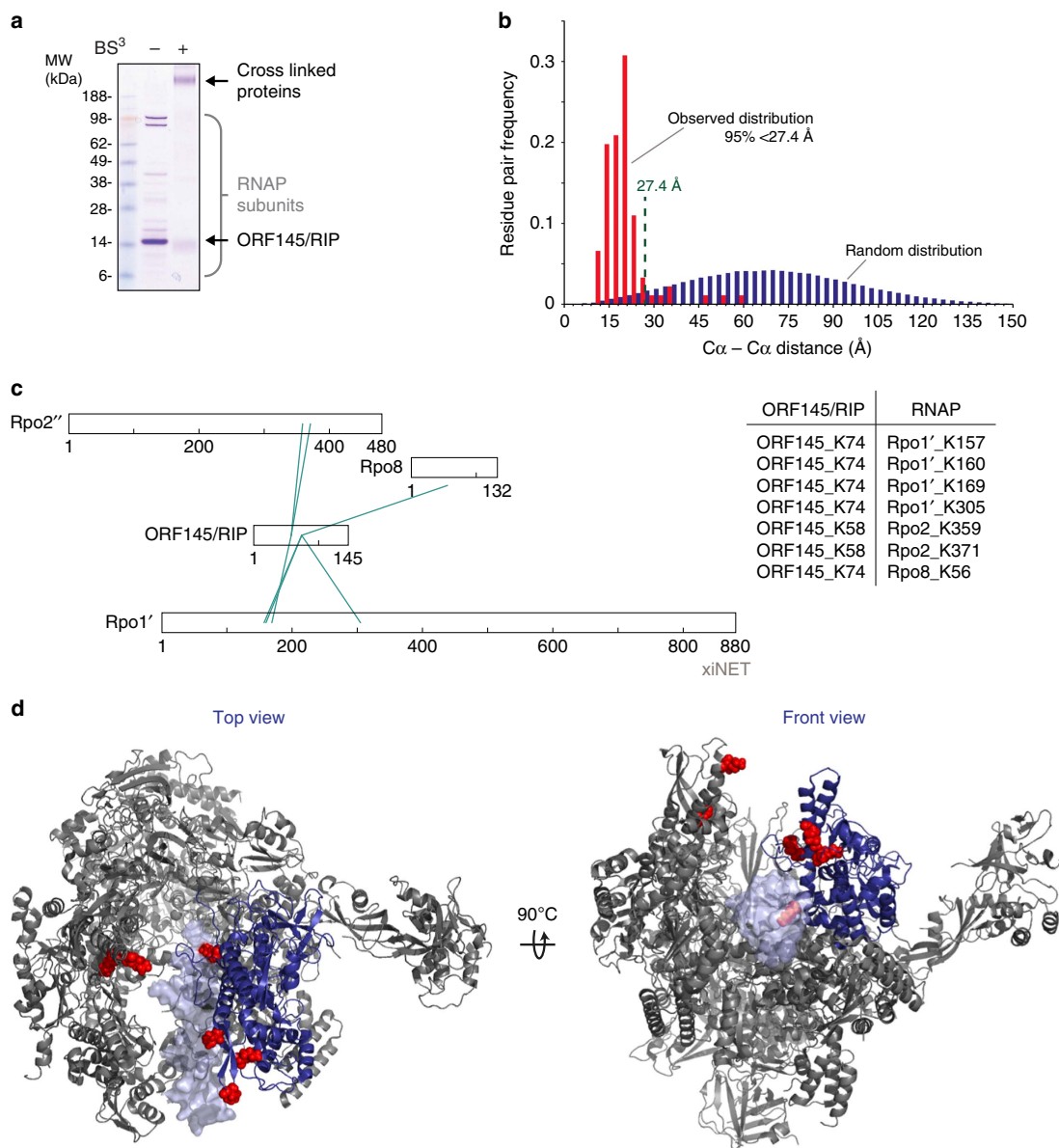

**Figure 2 | ORF145/RIP binds in the DNA-binding channel of RNAP.** (**a**) Chemical crosslinking of ORF145/RIP to the Sso RNAP. SDS–PAGE of Sso RNAP and ORF145/RIP before and after BS3 treatment. (**b**) Distance distribution of Lys–Lys links mapped on the RNAP structure (red bars) compared with a random distribution (blue bars) reveals that >95% of links are below the limit of the BS3 (Cα–Cα 27.4 Å). (**c**) Crosslinking network between RNAP and ORF145/RIP using XiNet[66]. The table summarizes the amino-acid residues of ORF145/RIP and RNAP subunits that are crosslinked by BS3. (**d**) BS3-reactive lysine residues are shown on the structure of the Sso RNAP–DNA complex (pdb 4B1O, front and top views), crosslinked lysine residues are highlighted as red spheres, the RNAP clamp in blue and the double-stranded DNA template light blue.

were analysed using mass spectrometry. In total, 88 crosslinked residue pairs were identified at 5% false discovery rate, of which the majority fell within RNAP. The intra-RNAP crosslinks were used to validate the data by verifying known RNAP subunit interactions (Supplementary Fig. 2A). Furthermore, a comparison of the C–α distance distribution for experimentally observed residue pairs and a random distribution within the RNAP ascertain the high quality of the data (Fig. 2b). The approximate crosslink limit for BS3 is 27.4 Å (maximum C–α distance); the vast majority of observable residue pairs (71 residue pairs) fall below this limit and are in good agreement with the X-ray structure of the Sso RNAP (pdb 4B1O). We detected six crosslinks between ORF145/RIP and RNAP (Fig. 2c) that were structurally consistent with each other (Fig. 2d and Supplementary Fig. 2B). Four of these crosslinks mapped to the

inside of the clamp domain, thus corroborating the results of the far-western and the SEC. The two further crosslinks map to the Rpo2 lobe that is positioned opposite the clamp in the DNA-binding channel (Fig. 2d). An additional single crosslink was observed between ORF145/RIP and Rpo8 (Fig. 2c), which is located distal to all other crosslinks and is likely an artefact of the crosslinking reaction. In summary, ORF145/RIP binds in the RNAP–DNA-binding channel between the clamp and protrusion motifs, and suggests that it may sterically occlude DNA binding.

**Inhibition of PIC formation**. To assess how binding of ORF145/RIP within the DNA-binding channel may affect the formation of DNA–TBP–TFB–TFE–RNAP transcription PIC, we performed EMSA experiments using $^{32}$P-labelled SSV1 T6 promoter

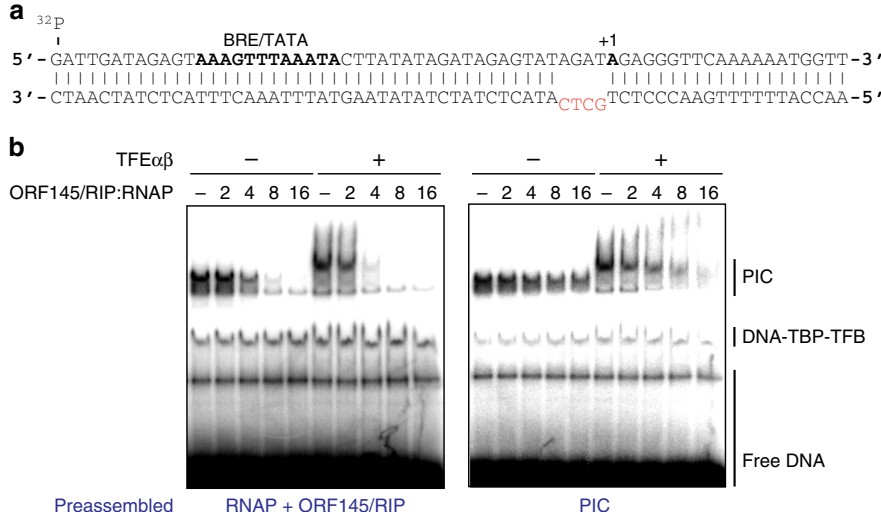

**Figure 3 | ORF145/RIP inhibits PIC formation.** (**a**) Sequence of the SSV1 T6 promoter used in PIC formation. The BRE/TATA motif and the $+1$ position are highlighted in bold and the non-complementary $-4$ to $-1$ regions are shown in red. (**b**) EMSA monitoring the formation of the archaeal PICs. Minimal archaeal PICs contain promoter DNA, TBP, TFB and RNAP. The addition of ORF145/RIP inhibits the formation of PICs in a dose–response dependent manner. The inclusion of TFEαβ leads to a supershift of the minimal PIC, which remains sensitive to ORF145/RIP. In the left panel RNAP and ORF145/RIP were incubated prior to the addition to DNA–TBP–TFB, while in the right panel PICs were allowed to form prior to the addition of ORF145/RIP.

templates (Fig. 3). In EMSAs the recruitment of Sso RNAP to the promoter is strictly dependent on the basal factors TBP and TFB, while TFE increases the stability of the PIC[11]. Since closed PICs are inherently unstable in Sso, we used a pre-melted template with four non-complementary base pairs at positions $-4$ to $-1$. Note, however, that this template does not alter the requirement for both TBP and TFB in RNAP recruitment (Fig. 3a and Supplementary Fig. 3A)[11]. The addition of ORF145/RIP decreases the PIC signal in a dose–response manner both in the absence and presence of TFE. Moreover, almost all PICs have disappeared at an eightfold molar excess of ORF145/RIP (Fig. 3b). In order to further elucidate the competitive nature of the ORF145/RIP inhibition, we reversed the order of addition of the components in the EMSAs. If PICs were allowed to form prior to the addition of ORF145/RIP, a higher molar excess of ORF145/RIP was required to obtain efficient inhibition, and even at 16-fold excess a significant fraction of PICs could still be observed (Fig. 3b). TFE binds to the outside of the RNAP clamp, and in the presence of TFE lower ratios of ORF145/RIP to RNAP were required to inhibit PIC formation. This suggests that TFE-bound RNAP increases the apparent affinity of ORF145/RIP, which could be caused by TFE's clamp-opening activity[14]. Native gel-binding experiments did indeed reveal that TFE subtly increases the binding affinity between ORF145/RIP and RNAP (Supplementary Fig. 3B). In summary, ORF145/RIP interferes with the formation and/or stability of the PIC.

**ORF145/RIP locks the RNAP clamp in a singular fixed position.** Considering its binding site within the DNA-binding channel, we explored the impact of ORF145/RIP on the conformation of the clamp. We have recently developed a single-molecule Förster Resonance Energy Transfer (FRET) assay that can monitor distance changes between a fluorescent donor–acceptor dye pair engineered into two residues that are located opposite of each other across the DNA-binding channel (Fig. 4a, Rpo1′-257 and Rpo2″-373 are highlighted as green and red spheres, respectively)[14]. Recombinant 12-subunit RNAPs were assembled from the two fluorescently labelled Rpo1′ and Rpo2″ subunits,

a biotinylated variant of Rpo11 for immobilization and wild-type variants of the remaining 10 subunits. Labelled RNAPs are able to engage with general transcription factors and are catalytically active. Our set-up uses the archaeal Mja RNAP, which interacts with ORF145/RIP to the same extent as Sso (Fig. 1c and see below). In order to test whether ORF145/RIP binding modulates the position of the RNAP clamp, we compared the FRET efficiency distribution of RNAP with RNAP–ORF145/RIP complexes. In a previous study the immobilized RNAP shows two FRET populations with a major high FRET efficiency $(E) = 0.67 \pm 0.01$ $(82 \pm 4\%)$ corresponding to an inter-dye distance of 46 Å and a minor low FRET population at $E = 0.40 \pm 0.03$ $(18 \pm 3\%)$ corresponding to 56 Å. These can be assigned to a RNAP clamp in a closed and an open conformation, respectively (Fig. 4b)[14]. In contrast, the RNAP–ORF145/RIP complexes are radically different with only a single population of intermediate FRET efficiency $(E = 0.58 \pm 0.01$; Fig. 4b), corresponding to 49 Å. Control experiments measuring fluorescence anisotropy of the donor and acceptor dyes ruled out that these changes are due to restricted mobility of the fluorescent dyes (Supplementary Table 1). These results show that ORF145/RIP disfavours both of the naturally occurring clamp conformations and instead locks the clamp into a single, intermediate conformation. The inability of the clamp to adopt any of the two conformations of the RNAP suggests that ORF145/RIP interferes with the conformational flexibility of the RNAP clamp that is essential for RNAP activity.

**Inhibition of transcription from host and viral promoters.** To determine the effect of ORF145/RIP on RNAP activity, we used a range of *in vitro* transcription assays designed to dissect the individual stages of transcription. First, we applied abortive transcription assays[11] using the well-characterized SSV1 T6 model promoter and the host Sso rRNA promoter (Fig. 5a). In this assay the promoter-bound RNAP extends a primer dinucleotide substrate by a single NTP. Synthesis of the trinucleotide product is strictly dependent on the promoter sequence, the dinucleotide and NTP[11], and the two factors TBP

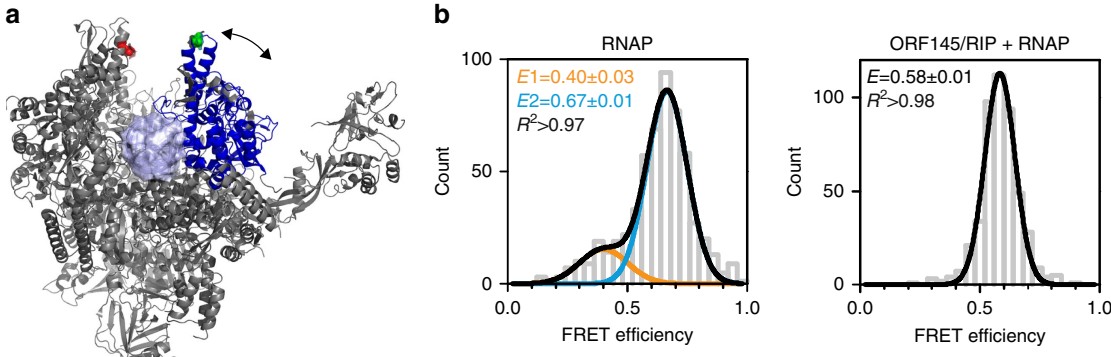

**Figure 4 | Single-molecule FRET analysis reveals that ORF145/RIP locks the RNAP clamp in a defined conformation.** (**a**) RNAPs were double-labelled with a FRET donor–acceptor dye pair (DyLight500-Dylight650) at amino-acid residues Rpo1′-E257 and Rpo2′′-Q373 indicated as green and red spheres, respectively (pdb 4B1O, the mobile clamp is shown in blue). (**b**) Single-molecule FRET histograms of RNAP (left)[14] and ORF145/RIP–RNAP complexes (right). The RNAP clamp exists in open and closed conformations represented as a low ($E = 0.40 \pm 0.03$, fit with a Gaussian function shown in orange) and high ($E = 0.67 \pm 0.01$, fit with a Gaussian shown in blue) FRET population, respectively. In contrast, the ORF145/RIP–RNAP complex exhibits a single FRET distribution with an intermediate conformation ($E = 0.58 \pm 0.01$) that was fitted with a single Gaussian function. The mean FRET efficiencies ($E$) and the coefficient of determination ($R^2$) are given with s.e.'s in the histograms.

and TFB, while the addition of TFE enhanced the signal. The addition of ORF145/RIP efficiently inhibited the formation of the first phosphodiester bond from both T6 and rRNA promoters in a dose-dependent manner (Fig. 5a).

This led us to investigate whether ATV promoters would also be subject to repression. The transcription start sites (TSSs) of ATV have not been mapped experimentally; however, the majority of mRNAs from its host Sso are leaderless, with the TSS within 5 bp of the translation start site[31]. We aligned the genomic regions proximal to the start codons of ATV in order to search for conserved sequence features and found an AT-rich region with a spacing relative to the start codon that is very similar to the TATA boxes from host Sso promoters[31]. This indicates that leaderless mRNAs are also a dominant feature of the ATV transcriptome (Supplementary Fig. 4A). Searching for specific sequence motifs in the region proximal to the start codons ($-100$ to $+50$ bp) identified a probable TATA element (TTTWWAA) in 26 of the 46 predicted promoter regions with a distance of $25 \pm 5$ bp from the start codon (Supplementary Fig. 4B). Interestingly, no sequence motif comparable to the BRE (A-rich) was identified in this search, which hints at subtle differences between virus and host promoters. On the basis of this information, we designed DNA templates for transcription assays containing the predicted promoters of ATV Gp63 and of ATV Gp48 and a 4-bp region of non-complementarity (see above; Supplementary Fig. 4C)[6,11]. Both ATV Gp63 and ATV Gp48 promoters facilitated transcription in a strictly TBP- and TFB-dependent manner and were stimulated by TFE, which validated our promoter prediction (Fig. 5a). Importantly, the addition of ORF145/RIP efficiently repressed the activity of both ATV promoters in this system at similar concentrations to the T6 and rRNA promoters (Fig. 5a). When comparing the inhibitory effect of ORF145/RIP in the presence and absence of TFE, ORF145/RIP inhibited at lower concentrations in the presence of TFE, in line with our observations from the EMSA experiments (Fig. 5a and Supplementary Fig. 4D). In summary, ORF145/RIP inhibits transcription from host as well as virus promoters in agreement with its ability to directly target the RNAP.

**Inhibition of productive stages of transcription.** To investigate the effect of ORF145/RIP on productive transcription initiation, we carried out transcription run-off assays using the strong SSV1 T6 and Sso rRNA promoters (Fig. 5b). Transcription from these

promoters generates 68 and 50 nucleotide run-off products, respectively, in a TBP- and TFB-dependent manner. The addition of ORF145/RIP inhibits transcription from both promoters in a dose-dependent manner (Fig. 5b). Moreover, ORF145/RIP is also capable of inhibiting SSV1 T6-directed transcription by the Mja RNAP (Supplementary Fig. 4E), further corroborating the view that ORF145/RIP binds to a highly conserved region of Rpo1′.

The elongation stage of transcription can be monitored independently of promoter elements and initiation factors using synthetic elongation scaffolds made of DNA and RNA oligonucleotides[32]. ORF145/RIP also represses transcription elongation in these assays (Fig. 5c) albeit to a lesser extent than run-off or abortive transcription. The binding site of ORF145/RIP indicates that it could prevent productive interactions between the RNAP and the initiation factor TFB and/or the DNA template in the DNA-binding channel. ORF145/RIP displays a similar order of addition dependence effect as that observed in the EMSA experiments (Fig. 5c), whereby the protein has a reduced effect if a transcriptionally competent elongation complex has already formed. As transcription elongation is not dependent on TFB, these results suggest that ORF145/RIP inhibition could be due to competition between ORF145/RIP and template DNA binding to the RNAP during transcription elongation. Alternatively, ORF145/RIP could inhibit elongation via an allosteric mechanism, for example, by translating conformational changes via the clamp into the RNAP-active site, or by preventing DNA movement along the DNA-binding channel. EMSA competition experiments revealed that ORF145/RIP and DNA/RNA elongation scaffolds do not compete for binding to RNAP (Supplementary Fig. 4F,G). In fact, supershift experiments using radiolabelled DNA/RNA elongation scaffolds and ORF145/RIP support an allosteric mechanism. When ORF145/RIP is added to RNAP–DNA/RNA complexes, a new complex with lower mobility is formed in a dose-dependent manner (Fig. 5d).

Together, these results demonstrate that ORF145/RIP binds directly to the RNAP and functions as an inhibitor of transcription. On the basis of its properties we rename ORF145/RIP as RIP.

**ORF145/RIP adopts an entirely α-helical fold.** To understand the structural basis for ORF145/RIP repression of RNAP activity, we generated a homology-based model of the protein using the I-TASSER and Phyre2 algorithms. The model is primarily based

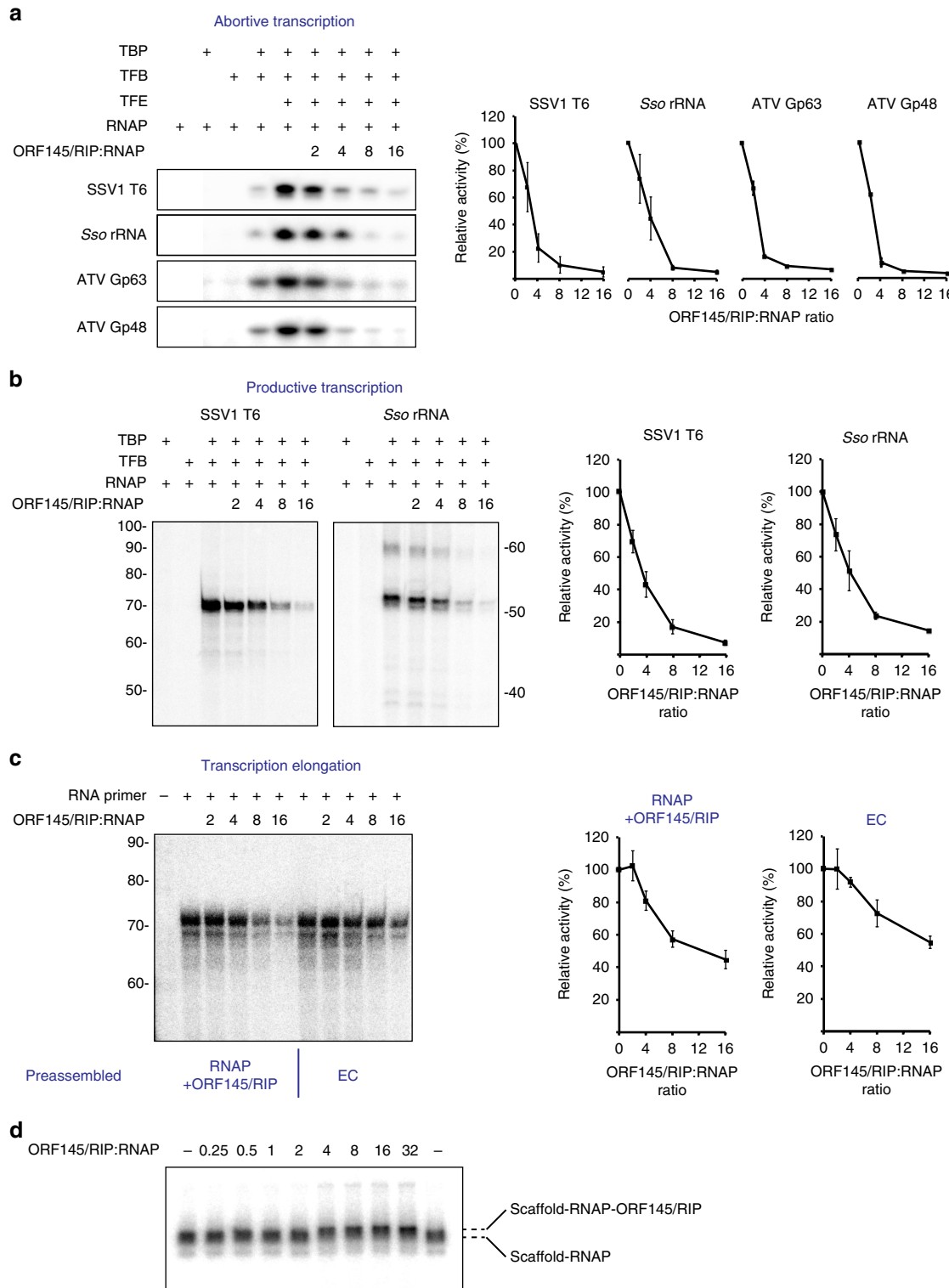

**Figure 5 | ORF145/RIP is a potent inhibitor of transcription.** (**a**) Abortive initiation assay using the SSV1 T6 promoter, Sso rRNA promoter, ATV Gp63 and ATV Gp48 promoters. The addition of ORF145/RIP represses the synthesis of a trinucleotide product in a dose-dependent manner. (**b**) Promoter-directed transcription initiation from all promoters is strictly dependent on TBP and TFB; ORF145/RIP efficiently represses transcription from both promoters in a dose–response manner. Sso RNAP can utilize synthetic elongation scaffolds containing DNA template- and non-template strands and a short RNA primer. Transcription elongation is repressed by ORF145/RIP in a dose–response manner (**c**). The preassembled RNAP–DNA–RNA complex is slightly less sensitive to ORF145/RIP repression when compared with reactions where the ORF145/RIP–RNAP complex was allowed to form prior to the addition of DNA–RNA scaffold (**c**). In each assay the transcripts were quantified using phosphorimaging and plotted as a function of ORF145/RIP–RNAP stoichiometry (right hand graphs on panels **a–c**). (**d**) EMSA demonstrating the simultaneous binding of the DNA–RNA scaffold and ORF145/RIP to RNAP. The small but significant difference in mobility between the RNAP-scaffold and the RNAP-scaffold-ORF145/RIP complexes is indicated with dashed lines. Error bars represent standard deviation from three technical repeats.

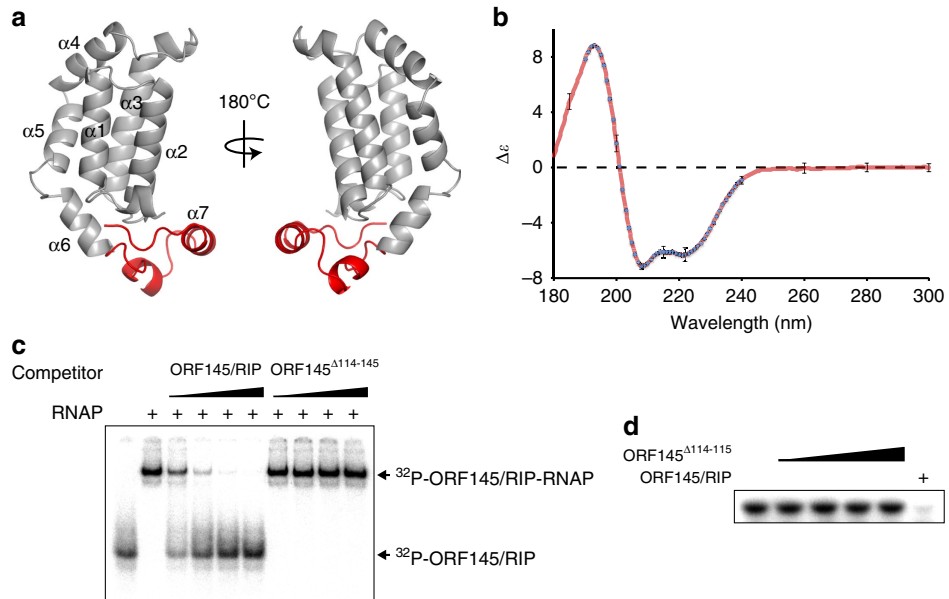

**Figure 6 | Structural homology model of ORF145/RIP. (a)** Homology model of ORF145/RIP prepared with I-TASSER (C-score − 2.12, TM-score 0.46 ± 0.15) shown as a cartoon representation with the C-terminal tail (114–145) coloured red. **(b)** CD spectrum of ORF145/RIP (red trace) confirms a very high alpha-helical content of the recombinant protein when compared with the spectra of proteins with known structures in Dichroweb (blue circles). **(c)** Truncation of the small C-terminal tail (Δ114–145) abrogates the binding of ORF145/RIP to RNAP. The EMSA shows that unlabelled wild-type ORF145/RIP competes for $^{32}$P-ORF145/RIP binding to RNAP, while ORF145$^{Δ114-145}$ does not. **(d)** ORF145$^{Δ114-145}$ does not inhibit abortive transcription. Error bars represent standard deviation of at least eight separate spectra.

on the parental structure of the ATV protein ORF131 (pdb 3FAJ) that, owing to its high abundancy in the virion particle, is predicted to be the major coat protein[33]. The similarity of the ORF131 and ORF145 sequences shows that they are derived from common ancestor and evolved by gene duplication and functional speciation[34]. The structural model predicts that ORF145/RIP is comprised entirely of α-helices, forming a tightly packed helix bundle core with a short C-terminal tail motif that protrudes at a right angle (Fig. 6a). The I-TASSER confidence scores for this model (C-score − 2.12, TM-Score 0.46 ± 0.15) are suggestive of a prediction with mid-range reliability. To evaluate this homology model experimentally, we used circular dichroism (CD) spectroscopy to characterize the secondary structure content of the protein. The CD spectrum displays a positive peak at 193 nm and two negative peaks at 208 and 222 nm, which are characteristic of a protein with predominantly α-helical content (Fig. 6b). Comparison of these spectra with those of proteins with known structures using Dichroweb allowed quantification of the secondary structure content. Results returned using different algorithms and data sets varied slightly. On the basis of the normalized root-mean-square deviations of the individual fits, visual inspection of the fitted spectra, the composition of the data sets and general consideration of protein structure, results suggest that ∼65% of amino acids adopt α-helical conformation, 11% are in turns and 23% disordered. Thus, the ORF145/RIP CD spectrum is in good agreement with the homology model. In order to investigate the role of the C-terminal tail in ORF145/RIP function, we created a short C-terminal truncation variant (deletion highlighted in red in Fig. 6a). The ORF145$^{Δ114–145}$ variant was expressed at high levels and proved to remain soluble after heat treatment, symptomatic of a correctly folded protein. This mutant was not able to compete with wild-type ORF145/RIP for the binding to RNAP (Fig. 6c) and, likewise, unable to repress transcription (Fig. 6d), which suggests that the C-terminal tail of ORF145/RIP is required for RNAP binding and thus inhibition.

**Overexpression of ORF145/RIP is detrimental to cell viability.** Our *in vitro* analysis of ORF145/RIP shows that it binds to RNAP with high affinity and efficiently represses all stages of transcription directed from host and viral promoters. These properties could lead to a global shut down of transcription *in vivo*. In order to test the effect of ORF145/RIP expression in the viral host, we developed an overexpression strategy using a plasmid-borne maltose-inducible system. The gene encoding ORF145/RIP was cloned into pSVA1450 that encodes the metabolic selection marker *pyrEF* before transforming into the uracil auxotrophic *S. acidocaldarius* strain MW001 (ref. 35). Transformations of a positive control vector, expressing the *lacS* gene, gave rise to 80 colony-forming unit (CFU) per μg ( ± 30 CFU per μg) on plates containing minimal growth substrate, whereas the ORF145/RIP overexpression plasmid did not yield any colonies from three independent experiments. To verify that the toxic effect resulted from the ORF145/RIP protein, we introduced a frameshift mutation after Met[5] into the *ORF145/RIP* gene, which produced 130 CFU per μg ( ± 75 CFU per μg) uracil autotrophic colonies, comparable to the *lacS* vector control. These results demonstrate that even under non-inducing conditions (in the absence of maltose) the small amounts of (leaky) ORF145/RIP expression have detrimental effects on the cell viability.

In summary, the expression of ORF145/RIP is highly toxic for the archaeal cell in good agreement with its predicted potential for repression of the cellular transcriptome.

## Discussion

We describe the first example of a virus-encoded protein that is able to bind directly to and repress the activity of the archaeal RNAP.

Despite being a small protein of only 145 residues, ORF145/RIP interacts with the archaeal Sso RNAP with very high affinity. Far-western blotting and crosslinking-mass spectrometry analysis reveal that ORF145/RIP binds within the DNA-binding channel

of RNAP in the proximity of the clamp and protrusion motifs on either side of the channel. This binding site provides important clues to the molecular mechanisms of ORF145/RIP function. In the context of the PIC, ORF145/RIP could interfere with the binding of transcription factor TFB core domain or DNA within the DNA-binding channel of RNAP, both would interfere with the PIC. EMSA experiments mimicking the transcription elongation complex demonstrate that ORF145/RIP can bind to RNAP in the context of the RNAP–DNA–RNA complex without disrupting it, while transcription elongation assays show that ORF145/RIP efficiently inhibits elongation. This supports the notion that ORF145/RIP binding to RNAP induces conformational changes that renders it inactive (Fig. 7a). The RNAP clamp switches between open and closed conformations as it progresses through the transcription cycle; during initiation the clamp opens to allow DNA loading and then closes to form a stable and processive transcription elongation complex[14]. Our single-molecule FRET analysis shows that ORF145/RIP perturbs the conformation of the clamp. ORF145/RIP might act like a wedge in the DNA-binding channel locking the clamp into one singular state that is likely to prevent 'mission critical' structural changes of the RNAP clamp. Several antibiotics inhibit bacterial RNAP by preventing movements of the clamp with a mechanism referred to as hinge-jamming, which demonstrates that the flexibility of the clamp is key to RNAP function[36,37]. We have prepared a structural homology model of ORF145/RIP and verified it by assessing the secondary structure content using CD spectroscopy.

The model reveals that the width of ORF145/RIP (~20 Å) is compatible with its insertion into the DNA-binding channel of RNAP; it is a 'tight fit' in line with ORF145/RIP binding negating the closed clamp state. The outcome of these steric and/or allosteric mechanisms is that ORF145/RIP is a potent repressor of all productive stages of transcription *in vitro*. These include promoter-directed formation of the first phosphodiester bond during abortive initiation and productive initiation, as well as promoter- and factor-independent transcription from elongation scaffolds.

Since ORF145/RIP interacts with RNAP independently of DNA and represses host as well as ATV promoters, its inhibitory activity is predicted to be global *in vivo*. In agreement with this, *S. acidocaldarius* cells are extremely sensitive to the ectopic expression of ORF145/RIP. In a similar experiment, transformation of a plasmid carrying the ORF145/RIP gene into an Sso strain that carries a CRISPR spacer targeting ORF145/RIP was extremely inefficient[38]. Sequencing of these colonies revealed mutations within the ORF145/RIP promoter region and inactivation of the CRISPR systems, suggesting that expression of ORF145/RIP is toxic to the cells[38]. Since ORF145/RIP is an abundant component of the ATV virion[6], transcriptional repression can take effect immediately following infection, and the repression by ORF145/RIP could help protect transcription of viral DNA and viral transcripts from being prematurely degraded by the host type III-B CRISPR system before and after the viral genome has been integrated into the protective environment of

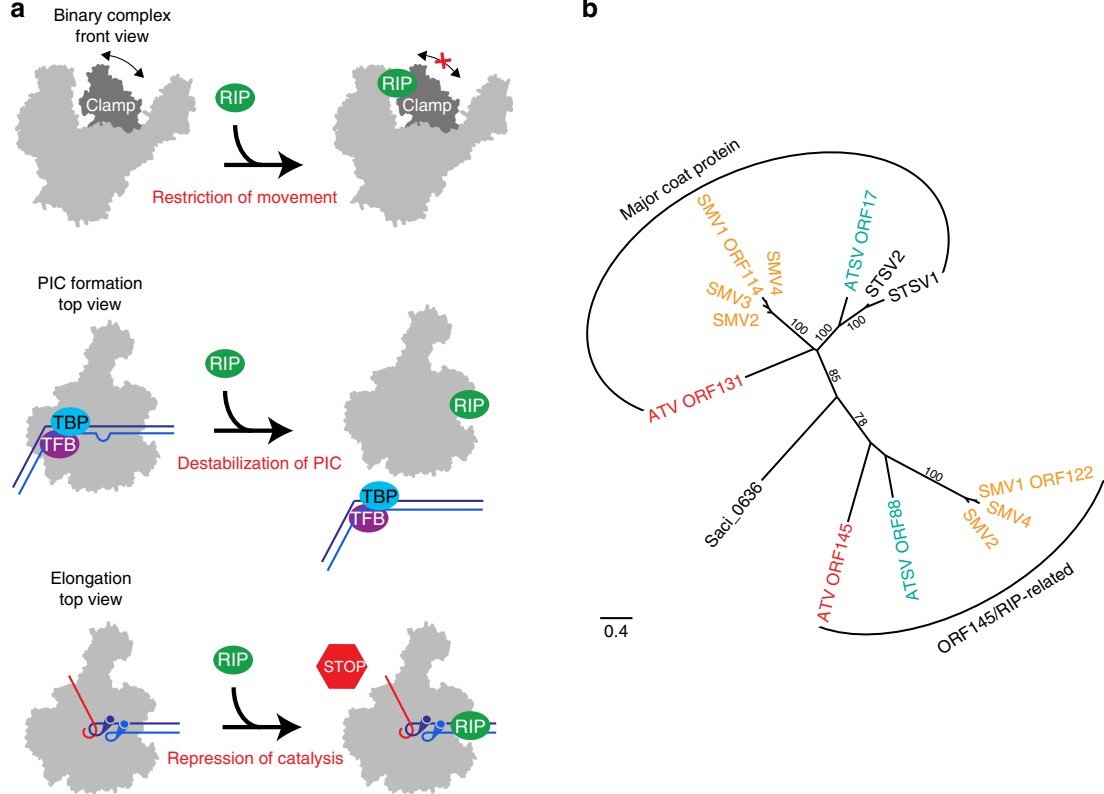

**Figure 7 | Molecular mechanism and evolution of ORF145/RIP.** (**a**) Schematic illustration of the molecular mechanism of RNAP repression. ORF145/RIP binds in the DNA-binding channel of RNAP and restricts the movement of the RNAP clamp. This destabilizes PIC complexes, and thus inhibits abortive and productive transcription. ORF145/RIP binds to RNAP simultaneously with the DNA/RNA template resulting in a repressed form of the transcription elongation complex. (**b**) The ORF145/RIP phylogenetic tree; maximum likelihood phylogenetic reconstruction of ATV ORF145/RIP and ORF131 and related proteins. Phylogenetic analysis suggests that the viral proteins fall into two different clades: ORF145/RIP-related and ORF131-related (major coat proteins). The scale bar represents amino-acid changes per site. Bootstrap values (100 replicates) are shown next to each branch. An alignment of ATV_ORF131 and ORF145/RIP-related proteins is available in Supplementary Fig. 5.

the host genome. The intracellular ratio between ORF145/RIP and RNAP as well as the half-life of ORF145/RIP are likely to play pivotal roles in setting the level of transcriptional repression.

The function of ORF145/RIP—apparently global repression of the transcriptome—has not previously been described for any factor encoded by an archaeon or by archaeal viruses; however, it is reminiscent of phage regulators in bacteria. Factors like T7 gp2 inactivate the host gene expression machinery by inhibiting the host RNAP to exploit remaining host resources and to allow efficient transcription of the phage genome. However, this strategy is more plausible if the virus provides an alternative viral RNAP that is resistant to the repressor. About 150 archaeal viruses have been sequenced to date and none of these—including ATV—encode a detectable viral RNAP (Eugene Koonin—personal communication, and ref. 39). In addition, the similarity of ATV and host promoter motifs—in particular the TATA element—reveals that ATV utilizes host initiation factors and RNAP (Supplementary Fig. 4A,B). Our results show that ORF145/RIP represses host as well as ATV promoters. We cannot rule out that as yet unidentified ATV-encoded transcription factors differentially modulate ORF145/RIP repression of host and ATV promoters. ATV includes two putative transcription factors (ATV gp28 and 29) with $Cys_2$-$His_2$ Zn-ribbon domains[39], and it is possible that factors like gp28 and 29 enable ATV promoter expression in the presence of ORF145/RIP, even though our sequence analysis of ATV promoters did not reveal any obvious putative binding sites.

Sequence homology searches reveal that ORF145/RIP belongs to a growing family of proteins encoded by ATV, *Sulfolobus* monocaudavirus (SMV) and *Acidianus* tailed spindle virus (ATSV) viruses and even includes a relative in the cellular genome of *S. acidocaldarius*. A phylogenetic analysis reveals that this family can be divided into two main clades that we coin 'major coat protein' and 'ORF145/RIP-related' (Fig. 7b). The genomic location of the cellular ORF145/RIP homologue (SACI_RS03025) proximal to integrase and tRNA genes is suggestive of a viral origin of the gene. Two members of the family have been structurally characterized: ATV131, the 'parent' used to prepare the ORF145/RIP homology model and ATSV ORF17, the major coat proteins of ATV and another virus termed ATSV, respectively[33,40]. The region of homology encompasses the four-helix bundle resolved in the ATV131 structure; however, most members include additional C-terminal extensions that likely confer specific functional properties to the structurally conserved core of the protein (Fig. 6a and Supplementary Fig. 5). In this context, its important to note that the C-terminal deletion variant of ORF145/RIP completely abrogated RNAP binding and inhibition (Fig. 6c,d). Our phylogenetic analysis suggests that, while ORF145/RIP shares a common ancestry with the structural protein ATV ORF131 (Fig. 7b and Supplementary Fig. 5), its function has evolved later following duplication and speciation.

In conclusion, we present the first identification of a virally encoded global inhibitor of transcription in archaea that functions by targeting the host RNAP via an allosteric mechanism. In evolutionary terms, ORF145/RIP is one of the most striking examples of functional diversification of a viral protein into a global transcription repressor or major coat protein.

## Methods

**Plasmids and proteins**. All oligonucleotides used in this study are detailed in Supplementary Table 2. Expression plasmids encoding ATV_ORF145/RIP were constructed by cloning the *ORF145/RIP* gene followed by a protein kinase A recognition sequence and C-terminal His-tag, or a thrombin cleavage site followed by a C-terminal His-tag, into pET21a+ (Novagen). ORF145/RIP was expressed in *E. coli* and purified by heat treatment at 76 °C and nickel affinity chromatography; the column was equilibrated with HN250 (10 mM HEPES pH 8.0 and 250 mM NaCl) before loading of the cleared cell lysate and elution by HN250 with 250 mM

imidazole. Peak fractions were analysed by 14% Tris-tricine SDS–PAGE and pure fractions were combined and dialysed into HN250 before treatment with thrombin. The sample was re-purified using nickel affinity chromatography, with the cleaved protein eluting in the flow through. Pure fractions were snap-frozen and stored at −80 °C. RNAP was isolated from the Sso strain M16:pSVA158 (ref. 41) using nickel and heparin affinity chromatography[11] Sso TBP, TFB and TFEαβ were heterologously expressed in *E. coli*. Sso TBP was purified by heat treatment and heparin affinity chromatography. Sso TFB was purified by heat treatment, nickel affinity, heparin affinity and SEC[42]. Sso TFEαβ and the RNAP clamp (a recombinant fusion of Rpo2 1055–1117, Rpo1′ 4–315 and Rpo1″ 340–377) were purified using heat treatment and nickel affinity chromatography[41].

**SEC and Western blotting**. Sso RNAP (0.2 nmol) and ORF145/RIP (1 nmol) were incubated in TN150 (25 mM Tris pH 8.0, 150 mM NaCl, 10 mM $MgCl_2$, 100 μM $ZnSO_4$ and 10% glycerol) at 65 °C for 10 min and resolved on a Superose 6 column (GE Healthcare) pre-equilibrated with TN150. Peak fractions were resolved by 14% Tris-tricine SDS–PAGE, transferred to nitrocellulose membranes using a fully wet tank system for RNAP detection or semi-dry blotting system for ORF145/RIP detection. The blots were then blocked in TBS with 5% milk powder and washed twice with TBS plus 0.1% tween and once with TBS before immunodetection. The blots were incubated with their respective anti-sera; 1:1,000 diluted polyclonal rabbit anti-serum against recombinant ORF145/RIP (Davids Biotechnology) and 1:1,000 diluted rabbit anti-serum against Sso RpoB (obtained from Steve Bell, Indiana University, USA)[43]. After washing, the anti-ORF145/RIP blots were incubated with 1:2,500 diluted anti-rabbit horseradish peroxidase (Promega #W4011), washed again, incubated with femto-ECL (Thermo Scientific) and imaged using Li-cor Odyssey. The anti-Sso RNAP blots were incubated with 1:10,000 diluted Dylight 680-conjugated goat anti-rabbit IgG (Thermo Scientific #35568), washed and scanned on a Typhoon FLA 9500 scanner (GE Life Sciences) equipped with a 685 nm laser.

RNAP clamp and ORF145/RIP gel filtration experiments were carried out using a Superose 12 column (GE Healthcare) and TK150 running buffer (50 mM Tris/HCl pH 8.0, 150 mM KCl, 100 μM $ZnSO_4$ and 5 mM dithiothreitol (DTT)) and analysed by 14% Tris-tricine SDS–PAGE and SYPRO-Ruby (Bio-Rad) staining.

**EMSAs**. ORF145/RIP was labelled with $^{32}P$ using protein kinase A (NEB) as per the manufacturer's instructions. Binding reactions were conducted in 16 μl volumes in lo-bind microcentrifuge tubes (Eppendorf). $^{32}P$-ORF145/RIP (1 nM) was incubated with 1–100 nM Sso RNAP in 10 mM MOPS pH 6.5, 10 mM $MgCl_2$, 260 mM NaCl, 5 mM DTT, 10% glycerol, 0.067 mg $ml^{-1}$ BSA and 2 mM ATP at 65 °C for 5 min. Complexes were resolved on 6% native PAGE, imaged by phosphorimagery and quantified using ImageQuant TL software package (GE Life Sciences). The binding curve was fitted using the tight binding equation[44]: % Complex = {($K_d$ + [RNAP]$_T$ + [RIP]$_T$) − √(($K_d$ + [RNAP]$_T$ + [RIP]$_T$)$^2$ − 4[RNAP]$_T$[RIP]$_T$)}/2[RIP]$_T$.

SSV1 T6 promoter oligonucleotides were labelled and purified. Reactions were performed in 16 μl volumes containing 10 mM MOPS pH 6.5, 10 mM $MgCl_2$, 260 mM NaCl, 5 mM DTT, 10% glycerol, 0.067 mg $ml^{-1}$ BSA, 20 μg $ml^{-1}$ heparin, 1 μM Sso TBP, 0.125 μM Sso TFB (0.5 μM Sso TFE), 50 nM Sso RNAP, 7.8 nM $^{32}P$-labelled DNA probe and 100–800 nM ORF145/RIP. Samples were incubated at 65 °C for 5 min with either pre-incubation of RNAP–ORF145/RIP before further incubation with the ternary complex or pre-incubation of the PIC before addition of ORF145/RIP. Complexes were resolved with 4–12% native PAGE and imaged using phosphorimagery.

**Far-western blotting**. Far-western blotting was carried out identical to western blotting with an additional ORF145/RIP incubation step (0.5 μg $ml^{-1}$ ORF145/RIP in TN150) and wash steps prior to incubation with the primary polyclonal anti-serum raised against ORF145/RIP and then Dylight 680-conjugated goat anti-rabbit IgG (Thermo Scientific).

**Crosslinking/mass spectrometry**. Purified RNAP and ORF145/RIP were combined and 225 μg BS3 (Thermo Fisher Scientific) dissolved in 100 μl crosslink buffer (20 mM HEPES pH 7.8, 20 mM NaCl, 5 mM $MgCl_2$) was added. The reaction was incubated at room temperature for 1 h and stopped by adding 1 μl of 2.5 M ammonium bicarbonate for 30 min at room temperature. The reaction mixture was separated on a 4–12% SDS–PAGE and stained with Coomassie.

Crosslinked bands were excised and the proteins reduced/alkylated and digested using trypsin[45]. The equivalent of 20 μg were injected directly in six portions, 2 × 1 μg and 4 × 3 μg. Further fractionation of crosslinked RNAP–ORF145/RIP peptides was carried out on 60% of the crosslinked sample (30 μg) using SCX-StageTips[46] following the protocol for linear peptides[47], yielding three fractions, followed by desalting using self-made C18 StageTips[48].

Peptides were loaded directly on a spray-emitter analytical column (75 μm inner diameter, 8 μm opening, 250 mm length; New Objectives), packed with C18 material (ReproSil-Pur C18-AQ 3 μm; Dr Maisch GmbH, Ammerbuch-Entringen, Germany) using an air pressure pump (Proxeon Biosystems)[49] at a flow rate of 0.5 μl $min^{-1}$. Mobile phase A consisted of water and 0.1% formic acid. Mobile phase B consisted of 80/20 acetonitrile:water and 0.1% formic acid. Peptides

were eluted over 169 min using a linear gradient going from 2% B to 40% B, followed by an increase to 95% B over 11 min. Eluted peptides were sprayed directly into a hybrid linear ion trap—Orbitrap mass spectrometer (LTQ-Orbitrap Velos, Thermo Fisher Scientific). Peptides were analysed using a high/high strategy, detecting at high resolution in the Orbitrap and analysing the subsequent fragments also in the Orbitrap. Fourier transform mass spectrometry (FTMS) spectra were recorded at 100,000 resolution and the eight most intense signals in the survey scan for each acquisition cycle were isolated with an $m/z$ of 2 Th and fragmented with collision-induced dissociation in the ion trap. $1+$ and $2+$ ions were excluded from fragmentation. Fragmentation (MS2) spectra were acquired in the Orbitrap at 7500 resolution. Dynamic exclusion was set to 90 s and repeat count was 1. Mass spectrometry (MS) raw files were processed into peak lists using MaxQuant version 1.3.0.5 (ref. 50) using default parameters except the setting for 'top MS/MS peaks per 100 Da' being set to 100. Peak lists were searched using a database containing the sequences of the subunits of RNAP from Sso and the sequence of ORF145/RIP using Xi (ERI, Edinburgh) for identification of crosslinked peptides. Search parameters were MS accuracy, 6 p.p.m.; MS/MS accuracy, 20 p.p.m.; enzyme, trypsin; specificity, fully tryptic; allowed number of missed cleavages, four; crosslinker, BS3; fixed modifications, carbamidomethylation on cysteine; variable modifications, oxidation on methionine. Linkage specificity for BS3 was assumed to be at lysine, serine, threonine, tyrosine and protein N termini. Identified crosslinked residue pairs were returned at an estimated false discovery rate of 5%, which was carried out following a modified target-decoy search strategy[45,51].

The MS data have been deposited to the ProteomeXchangeConsortium via the PRIDE partner repository with the data set identifier PXD001692.

**Single-molecule FRET microscopy.** The double-labelled RNAP system and its application to monitor the clamp conformation is as follows. Briefly, measurements on immobilized donor–acceptor fluorophore-labelled RNAP molecules were carried out using a homebuilt PRISM-TIRF set-up based on an Olympus IX71 with alternating laser excitation[14,42,52]. All measurements were carried out at room temperature (21 °C) on a polyethylene glycol (PEG) surface attached to a flow chamber. Quartz slides were prepared according to ref. 53. A yellow laser (568 nm; Coherent Sapphire 100 mW) was used for excitation of the donor, and a red laser (639 nm; Toptica iBeam Smart 150 mW) was used for direct excitation of the acceptor. The fluorescence was collected by a $\times 60$ Olympus/1.20 NA. water-immersion objective and split by wavelength with a dichroic mirror (640 DCXR, Chroma Technology) into two detection channels that were further filtered with two bandpass filters (Semrock BrightLine 582/75 and Semrock Brightline 609/54) in the orange channel and one longpass filter (647 nm Semrock RazorEdge) in the near infrared detection range. Both detection channels were recorded with one EMCCD camera (Andor IXon X3, preGain 5.1, gain 250, frame rate 10 Hz) in a dualview configuration (TripleSplit, Cairn UK) and the acquired data were analysed by custom-made software based on LabVIEW 2012 64 bit (National Instruments). The resulting histograms were fitted either with a single or double Gaussian fit and the mean FRET efficiency and the s.e. was determined from the fit. ORF145/RIP–RNAP complexes (0.1 μM RNAP, 1 μM ORF145/RIP) were allowed to preform at 65 °C for 20 min prior to immobilization on the streptavidin-coated surface and subsequent microscopy.

**ATV promoter prediction.** The ATV genome sequence and gene annotations were downloaded from NCBI genomes database. Overall, 35 of the 72 ATV genes are organized into 12 operons. The 72 ATV genes are predicted to be organized into 46 transcription units based on the orientation and distance of upstream genes[6]. For the 46 genes that constitute the first cistron of these transcription units, the DNA sequence from $-100$ to $+50$ relative to the annotated start codon was extracted using BedTools[54] and alignments performed using WebLogo[55]. Sequence motifs were determined using MEME (Multiple Em for Motif Elicitation)[56].

**Promoter-directed abortive initiation assays.** Proteins were incubated in 16 μl volumes containing 10 mM MOPS pH 6.5, 10 mM MgCl$_2$, 100 mM KCl, 5 mM DTT, 10% glycerol, 0.067 mg ml$^{-1}$ BSA, 5 μg ml$^{-1}$ heparin, 1 μM Sso TBP, 0.125 μM Sso TFB (3 μM Sso TFE), 50 nM Sso RNAP and 100–800 nM ORF145/RIP. Transcription was initiated with the addition of 50 nM of dsDNA template, 250 μM dinucleotide primer (IBA), 50 μM [$\alpha$-$^{32}$P]-nucleotide and incubated at 65 °C for 15 min before the addition of formamide-loading dye. Samples were resolved on a 20% 7 M Urea mini PAGE, imaged by phosphorimagery and quantified using the ImageQuant TL software package (GE Life Sciences).

**Promoter-directed run-off transcription assays.** Reactions were performed in 16 μl volumes containing 10 mM MOPS pH 6.5, 10 mM MgCl$_2$, 260 mM NaCl, 5 mM DTT, 10% glycerol, 0.067 mg ml$^{-1}$ BSA, 40 μg ml$^{-1}$ heparin, 1 μM Sso TBP, 1 μM Sso TFB, 50 nM Sso RNAP, 500 ng T6/rRNA promoter plasmid DNA and 100–800 nM ORF145/RIP. Transcription was initiated with the addition of 0.67 mM AUG/ACG trinucleotide mix, 8.3 μM CTP/UTP and 75 pM [$\alpha$-$^{32}$P]-CTP/UTP nucleotide and incubated at 76 °C for 5 min before the addition of for-mamide-loading dye. Samples were resolved on a 10% 7 M Urea sequencing PAGE, imaged by phosphorimagery and quantified using the ImageQuant TL software package (GE Life Sciences).

**Transcription elongation assays.** Reactions were performed using conditions identical to the promoter-directed run-off transcription assays but without the initiation factors TBP, TFB and TFE and using an increased heparin concentration (2 mg ml$^{-1}$; ref. 32). The synthetic DNA–RNA elongation scaffold was formed using 3.2 μM template strand, 5 μM non-template strand and 16 μM RNA.

**CD spectroscopy.** CD spectra were recorded at 0.34 mg ml$^{-1}$ of ORF145/RIP and a path length of in 0.2 mm in 50 mM sodium phosphate buffer, pH 8.0 with 150 mM NaF on a Jasco J-720 CD spectrometer. At least eight spectra were recorded from 300 to 180 nm, as were blank spectra of buffer. Spectra were quality-checked by overlaying and then averaged; the averaged buffer spectrum was then subtracted from the averaged ORF145/RIP spectrum. The resulting ORF145/RIP spectrum was zeroed on the baseline from 263 to 300 nm and converted to delta epsilon using its accurate concentration measured by quantitative amino-acid analysis. Secondary structure analysis was performed by submitting the spectrum to Dichroweb[57] using Data set 7 (ref. 58) and Data set SP175 (ref. 59) and the algorithms CDSSTR[60], Contin-LL[61] and Selcon 3 (ref. 62).

**Sequence alignments and phylogenetic analysis.** ORF145/RIP-related proteins were identified using PSI-BLAST[63] using a reciprocal approach that identified identical sets of homologous genes. Sequences were aligned using MUSCLE[64]. Maximum likelihood phylogenetic analysis was carried out using PhyML (http://www.atgc-montpellier.fr/phyml-sms/)[65] with standard settings and automated model selection by SMS (LG $+$ G6 $+$ I $+$ F) using BioNJ for the starting tree. In all, 100 bootstrap replicates were performed.

**Data availability.** The authors declare that the data supporting the findings of this study are available within the paper and its Supplementary Information Files. In addition, the mass spectrometry data have been deposited to the ProteomeX-changeConsortium via the PRIDE partner repository with the data set identifier PXD001692. All other data are available from the authors upon reasonable request.

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

## Acknowledgements

We are extremely grateful to Eugene Koonin at the NCBI for searching the ATV genome for cryptic RNAPs and unannotated transcription factors, and for rewarding discussions about virus survival strategies. Lee Whitmore was very helpful with the interpretation of the CD spectra. We like to acknowledge all members of the RNAP (Werner) and small RNA (Arnvig) laboratories for helpful discussions, and in particular Kristine Bourke Arnvig for critical reading of the manuscript. We would furthermore like to acknowledge Sonja Verena-Albers for helpful discussions of Sso overexpression constructs. Work in the UCL RNAP laboratory is funded by a Wellcome Trust Investigator Award to F.W. (096553/Z/11/Z), work in the Grohmann team is funded by the Deutsche Forschungsgemeinschaft (SFB960), the Rappsilber group is funded by a Wellcome Trust Senior Research Fellowship 103139, Centre core grant 092076 and instrument grant 108504.

## Author contributions

Conception of the study: F.W., C.S. and R.G.; experimental work: C.S., F.B., A.B. (mass spectrometry), S.S. (smFRET), T.D. (CD spectroscopy), K.S. (computational analysis—promoter prediction), E.M. (experimental contribution); data analysis: C.S., F.W., T.D., D.G. and J.R.; provided materials and equipment: S.E. and P.T.; writing of the manuscript: F.W., C.S., F.B., D.G. and J.R.

## Additional information

**Competing financial interests:** The authors declare no competing financial interests.

