## [Peer Review File · Nature Communications]

Reviewers' comments:

Reviewer #1 (Remarks to the Author):

ORF145 (or RIP) is the very first example of an archaeo-viral regulator of the host RNAP and in this study Sheppard et al provides a comprehensive biochemical characterisation of this protein. This study is therefore clearly novel and would be of substantial interest to researchers affiliated with the field of regulation of the transcription apparatus. The manuscript is clearly written and the quality of presentation is excellent; however, some sections are occasionally verbose and would benefit from shortening/focusing. Although all the experiments have been carefully designed and conducted, I have the following questions and suggestions for improvement:

1. Far-western blotting is used to determine the RNAP subunits to which RIP interacts with. The authors claim that all 13 subunits of the RNAP were tested for interaction with RIP. This data should be provided (at least as supplementary data). Further, later in the manuscript the authors claim that they can detect interactions between Rpo2 and Rpo8 and RIP (by MS). Did the Far-western experiments detect these interactions or was the MS data confirmed by Far-western blotting using Rpo2 and Rpo8 subunits?
2. The authors claim that they 'generated a synthetic recombinant RNAP clamp composed of three RNAP subunits' to detect the binding of RIP to the clamp. However, only a single subunit is seen in the SDS-PAGE analysis of the GF samples. Shouldn't there be three? Please clarify.
3. It is suggested that RIP binds more tightly to the RNAP-TFE complex than to the RNAP in the absence of TFE. Presumably, the affinity of RIP to RNAP and RNAP-TFE complex can be quantified by FA analysis? Also, can RIP interact with TFE? What is the evidence that it doesn't?
4. The section dealing with the in vitro transcription assays will benefit from substantial shortening.
5. If RIP affects the half-life of the PIC (as suggested) then this should be experimentally investigated.
6. Does RIP bind to the RNAP bound to an elongation scaffold? Presumably not or poorly, if the claim that RIP competes with template DNA (page 18, line 4) is correct?
7. The authors can experimentally demonstrate that RIP and DNA compete for clamp interaction by conducting an FA assay with labelled DNA and look for displacement of the DNA as a function of RIP concentration/time.
8. As presented, I did find that the section containing the homology modelling data of RIP adds little to advance our understanding of RIP function. The authors propose that the acidic patch might be functionally important considering that it binds to the clamp (=DNA binding channel). The authors are urged to conduct a mutational analysis of this patch or of any other region of functional significance) to further validate their claims of how RIP binds to and inhibits the RNAP. Clearly, what is required is the structure of RIP - but this is well beyond the scope of the manuscript and it would be unfair to ask the authors for it.
9. If RIP or RNAP is purified from *Sulfolobus* (under conditions where RIP is expressed and exhibits toxicity) - does RIP co-elute with the RNAP (or vice versa) and is the co-eluted RNAP composition/activity different in the presence/absence of RIP?

Reviewer #2 (Remarks to the Author):

I enjoyed reading this interesting paper. Sheppard et al. describes the mechanism of action of a viral protein from the archaeal virus ATV (called RIP by the authors) in repressing the archaeal DNA directed RNA polymerase. The manuscript unfolds nicely.

The paper includes an experimental part using classical biochemistry, FRET and CD, and a theoretical part using homology modeling.

The experimental part argues that the RIP protein binds to the clamp occluding the DNA entry channel impeding the loading of the DNA template onto the RNAP. Thus it is a potent inhibitor of transcription. The modeling part makes the claim that this protein possesses a helix-bundle fold similar to the major capsid protein of ATV.

The work is original as this is the first example of a viral archaeal repressor acting directly on the RNAP however the mechanism of action proposed replicates similar mechanisms observed in bacteriophage. From the methodological point view the main techniques are largely used in the field.

Both the experimental and modeling parts left me with various questions and I would be interested in hearing the author's views on the points that I will raise below.

This work may be of some interest to the readership of Nature Communication, but that the authors have a chance to answer comments from me and other referees and that the authors be more specific about the interactions governing their proposed repression mechanism by RIP.

Specific comments:

Page. 4 - Line 50. The idea of three virospheres is misleading even within quotes. So far the virophere remains one; please see the International Committee on Taxonomy of Viruses.

Page 10. - Lines 173-176. Although others have previously generated rClamp from Pfu and visualize it in 3D by X-ray crystallography whether also Sso rClamp folds correctly is a plausible assumption. However, the Fig. 1D red trace (rClamp alone) doesn't show a symmetrical elution peak. In unfolding and re-folding of certain proteins the lack of a symmetrical peak with streaking tails is suggestive of protein not properly folded -

I wonder if this is the case here. Can the authors explain the reasons for these profiles? Also the peaks on the Figure 1D (elution profiles) do not seem centered respectively at 16 ml (red trace) and 14 ml (blue trace) as stated in the text but more likely to 14.2 ml (red trace) and 13 ml (blue trace) unless the authors used the gel below to define the 'peak' elution which would be a bit unusual. Also why not perform CD on the Sso rClamp to assess its secondary structure in solution? MALS analysis would also be beneficial here.

Page 10. - Line 181. 'at higher resolution' appears a bit too grand - the authors have further mapped the potential binding sites of RIP onto the RNAP but not identified the interacting residues - I would suggest removing it.

Moreover the section ORF145/RIP binds in the DNA-binding channel describes more the protocol used for the experiment rather than the results of the experiment per se. For example there is no mention within the text of which Lysines within RNAP and RIP have been found cross-linked. The re-direction to the figures helps partially as those on the RNAP have been displayed in red. Also the order of presentation of Figures within the text doesn't follow the A, B, C but the A, C, B panels which is illogical.

The authors on page 12 lines 209-210 explain the observed binding to Rpo8 as possibly due to the

high protein concentrations. The question is: do the authors 'mix & match' RNAP and RIP for the cross-linking experiment or do they use the RNAP-RIP complex eluting from the SEC experiment? In the former case possible experiment duplicates at lower concentrations of RIP might untangle the described artifact; in the latter case no 'high protein concentration' issue should emerge.

Fig 2C would benefit from being a larger size, in particular the network involving RIP with Rpo8 and Rpo1' with marking of the Lys residues involved. The remaining network to the opinion of this reviewer is a bit confusing and it can be displayed as Supplementary Figure. Finally there is no mention of the fact that RIP has 14 lysines and of these, only 5 cross-link and they seem to be centrally located in the sequence.

Where would these lysines map onto the modeled RIP structure in Fig. 6A?

Page 13. - Line 236. Have the authors tested the possibility of RIP directly binding to DNA?

Page 14. - Line 257. The majority of the assembled RNAP in solution appear to have the clamp in closed conformation as the distribution is 82% (close) vs 18% (open). Thus RIP would bind to both conformations and lead to an intermediate conformation - although in 80% of the cases this intermediated conformation would be reflecting the RIP binding to the closed RNAP - how much is the distance between the donor and acceptor in the intermediate conformation? And how does it compare in terms of distances with the close and open ones? Have I missed this information within the text?

Also I was puzzled by the fact that Fig. 4B left appears to be the same as that below published by the same authors in reference 14 [*FIGURE REDACTED*] except for some aesthetic differences. Am I right?

There is nothing wrong in this as at page 14 lines 255-259 the reference 14 is indeed mentioned although the sentence would benefit from adding:

"In a previous study the immobilized RNAP....., respectively (Figure 4B) 14."

However, in the caption of Figure 4B - this is not mentioned - and it should be explicitly stated that Fig. 4B, left has been adapted from Reference 14.

Page 18 - Lines 335-338. These lines condense the essential findings of the manuscript, and are strongly supported experimentally.

Page 18 - Line 340 onwards. I am a bit confused in this section. The experimental data are the CD spectra whose deconvolution using algorithms in Dichroweb indicate that 65% adopt a-helical secondary structure. The homology model although suggestive remains a speculative model - a serious point is that there is no mention of how reliable the generated model is - do phyre2 and I-tasser have confidence indexes?

Also I would not go so far as to calculate the isopotential surface from a model extrapolated on a sequence identity of less than 30% - BTW how was the isopotential surface calculated? - there is no mention in the methods.

Moreover at page 18 lines 345-348 - the "It seems unlikely that.....similar functions, but their....." appears redundant as in reference 33 it is already stated that products of ORF131 and ORF145 are paralogs.

This entire section seems an add-on on the biochemical and FRET analysis and only CD data are reliable - I read this section several times and could not figure out what reliable insight the modelling

provides.

Page 21 - Lines 388 onwards. The discussion is clearly laid out but there are some lapses in assumptions.

At line 401 - 'RIP could act like.....' would be better 'RIP might act like...' as the evidence for the mechanism in vivo have not been ascertained. The evidence on how it would work comes from in vitro assembled RNAP whose 80% possess a closed clamp and 20% open. Difficult to extrapolate this observation to the transcription cycle. Also very little information is given on how this intermediate conformation would look like from smFRET data. Does 'intermediate' means between closed and open clamp? Again how much in ångström?

At lines 406-408: the experimental data on the RIP structure are the CD spectra, the rest is speculation, including the isopotential surface for the reasons above expressed.

The rest of the Discussion flows amenably and is based on the biochemical data gathered which appear indeed more solid and informative than the modeling and also somehow of the smFRET outcome. Moreover the sequence homology searches and the phylogentic tree of major capsid proteins and RIP-like proteins looks interesting but it proves that despite the sequence homology it is very hard to get insight into function and I would say structure as well.

What would have strengthened the manuscript is the identification of those residues onto RIP and/or RNAP that indeed interact. Do the authors know if is the N- or C- terminal tail of RIP that interacts with RNAP or the central body? Have they tried to assess by RIP truncation which portion of the protein is the interacting one? The cross-linking experiments although very useful do not directly provide this information.

The mechanism shown in Figure 7A is appealing for its simplicity but at the same time doesn't say much about specific interactions which would greatly help to understand at molecular level how the inhibition occur in the light also of the others transcription binding factors.

Thanks again for the chance to read this manuscript. This is undoubtedly well-presented and it reads amenably but there is a decided inbalance between the experimental results and how far these results have been interpreted.

Minor points:

-) Reference 14 was published in March 2015 - it should be updated.
-) Page 26 - line 504. A full stop should precede 'Sso TFE.....'
-) Page 33 - line 633: is it said why the '65 oC and 20 min' incubation conditions were used?

Reviewer #3 (Remarks to the Author):

The authors describe a protein of Acidianus two-tailed virus (ATV) that interacts with host RNA polymerase to inhibit transcription. They characterize this interaction using a variety of different strategies, all supporting the conclusion that the ORF145 protein binds to the DNA binding channel of RNAP, locking the flexible RNAP clamp into a fixed position. Once bound to RNAP, transcription of both host and virus promoters is blocked.

The study is significant in that it is the first description of a virus-encoded transcription factor that acts directly on host RNAP, rather than blocking its access to promoter sites. The manuscript was a pleasure to read, and I find the results of this multidisciplinary approach persuasive, and the data presented in a clear and concise manner. I believe the conclusions are valid, and I have only a few comments or suggestions that I think will improve the manuscript.

1. Lines 86/92. It would be useful to reference at least one of the early studies of gene regulation in archaeal viruses. I think the Ken and Hackett study of phiH repressor (Ken, R. & N.R. Hackett, (1991) *Halobacterium halobium* strains lysogenic for phage phi H contain a protein resembling coliphage repressors. *J Bacteriol* 173: 955-960.) is probably the earliest.
2. I was initially concerned that the 32P labelling of RIP might alter its binding properties, but the results obtained with this label fit in so well with data from unlabelled or fluor labeled RIP that I am persuaded that this was not an issue.
3. Line 291: I think 'indicated' is too strong a term given the lack of direct experimental proof at this stage. I would prefer 'suggested' or 'consistent with' or similar terminology.
4. Line 392: I don't think XL is defined before this, and assume you mean cross-linking. Either define this abbreviation when you first use the term or, better, just use cross-linking MS to make it clear.
5. In the discussion regarding levels of RIP and its influence on transcription, is there anything known about the transcription pattern of ORF145 or the half-life of RIP? For example, if it is only expressed at a specific time during infection, or if the protein degrades rapidly in cells. If there is published data on these aspects then they would be useful here.
6. Line 468: this describes an experimental study, and should not be part of the Discussion section. If the authors do not wish to integrate this into the results section, then I think it should be removed.
7. Line 672: I find this sentence difficult to follow, and it would be clearer if the component concentrations of the synthetic scaffold were stated more clearly. For example, the abbreviations TS-DNA and NTS-DNA do not seem to be given in full in the manuscript.
8. References: some attention needs to be taken with the references. Species names in italics (e.g. ref 16), and some variation in formatting (e.g. journal names in sentence or title case, such as refs 5 and 6).
9. Figure 1, panel C: I think it would be clearer to the reader if a bit more description of the Far-Western blot was given, including the antibody used for detection.
10. Figure 3, panel A: It would be clearer to me if the complementary bases of the two strands were indicated with vertical bars or dots between them. My initial impression looking at this sequence was that it was a single nucleotide sequence split over two lines. Indicating 5' and 3' ends might also be useful.
- 11: Line 1027: I think it is Phyre2 rather than a superscript 2.

Point-by-point response to reviewers comments

Blue denotes responses to reviewers comments or questions.

Red denotes changes that have been made to the text or figures.

Reviewers' comments:

Reviewer #1 (Remarks to the Author):

ORF145 (or RIP) is the very first example of an archaeo-viral regulator of the host RNAP and in this study Sheppard et al provides a comprehensive biochemical characterisation of this protein. This study is therefore clearly novel and would be of substantial interest to researchers affiliated with the field of regulation of the transcription apparatus. The manuscript is clearly written and the quality of presentation is excellent; however, some sections are occasionally verbose and would benefit from shortening/focusing. Although all the experiments have been carefully designed and conducted, I have the following questions and suggestions for improvement:

1. Far-western blotting is used to determine the RNAP subunits to which RIP interacts with. The authors claim that all 13 subunits of the RNAP were tested for interaction with RIP. This data should be provided (at least as supplementary data). Further, later in the manuscript the authors claim that they can detect interactions between Rpo2 and Rpo8 and RIP (by MS). Did the Far-western experiments detect these interactions or was the MS data confirmed by Far-western blotting using Rpo2 and Rpo8 subunits?

- Sso RNAP purified from archaeal biomass consists of 13 subunits. Only the Rpo1 subunits shows reactivity with RIP in Far-western blotting experiments using either highly purified Sso RNAP or Sso cellular extract. Both preps contain the full complement of 13 RNAP subunits and thus *all* subunits were tested for binding to RIP in our analysis
- The XL MS analysis provides evidence for proximity (distance between lysine residues less than 27.4Å) rather than direct binding, in contrast to Far-Western and the clamp-RIP SEC results which *do* reflect direct physical interactions. Our results show that Rpo1' binds to RIP, and that the Rpo1 clamp can form a stable complex with RIP, while the XL-MS not only support these interactions but furthermore suggests that RIP binds on the inside of the clamp – and thus proximal to the part of Rpo2 that is on the opposite side of the DNA binding channel.

2. The authors claim that they 'generated a synthetic recombinant RNAP clamp composed of three RNAP subunits' to detect the binding of RIP to the clamp. However, only a single subunit is seen in the SDS-PAGE analysis of the GF samples. Shouldn't there be three? Please clarify.

- The recombinant clamp is a synthetic fusion composed of fragments of three RNAP subunits (Rpo1', Rpo1'' and Rpo2) that are expressed and purified as 1 single polypeptide. We described this clamp construct in a paper published in 2015 (Elife; 4:e08378. doi: 10.7554/eLife.08378). *We have elaborated in the main text; lines 171-174: ...we generated a synthetic recombinant RNAP clamp composed of fragments of the Rpo1', Rpo1'' and Rpo2 subunits expressed as a single polypeptide chain^{11,28}.*

3. It is suggested that RIP binds more tightly to the RNAP-TFE complex than to the RNAP in the absence of TFE. Presumably, the affinity of RIP to RNAP and RNAP-

TFE complex can be quantified by FA analysis? Also, can RIP interact with TFE? What is the evidence that it doesn't?

- Fluorescent dyes often have a tendency to perform poorly at the elevated temperatures of our system. We have labeled RIP with Alexa 488 and Cy3B and attempted numerous fluorescence anisotropy (FA) experiments using a range of conditions – unfortunately the fluorophores were not stable at the elevated temperatures (65-76°C) required for – and biologically relevant to – the RNAP-RIP interaction. However, instead we have compared the binding of RNAP and RNAP-TFE complexes to RIP in EMSAs. As predicted, TFE subtly increase the affinity of RIP for RNAP. This increase in affinity is likely responsible for the increased ‘potency’ of RIP observed in the PIC EMSAs including TFE as compared to omitting TFE.
- These data have now been included as supplementary Figure S3B and mentioned in lines 237-239: *Native gel binding experiments did indeed reveal that TFE subtly increases the binding affinity between ORF145/RIP and RNAP (Figure S3B).*
- We have conducted several experiments to probe for a putative direct interaction between RIP and TFE – including EMSA. The results are always consistent and do not support that RIP and TFE form a complex. However, since the results are negative evidence we prefer not to include them in the manuscript.

4. The section dealing with the in vitro transcription assays will benefit from substantial shortening.

- Since we are well within the word limit of this type of manuscript, and since the other two reviewers do not express the same opinion, we would rather not crop the transcription section because we feel it would be less clear, and precise.

5. If RIP affects the half-life of the PIC (as suggested) then this should be experimentally investigated.

- Our order of addition EMSAs show that (i) the addition of RIP leads to the destabilization of a preformed PIC, and (ii) the inclusion of RIP prevents the formation of the PIC. We have been very careful in the manuscript not to claim that RIP affects the half life of the PIC, since this would require a kinetic analysis that we do not provide in any of the figures. Preliminary PIC EMSA experiments using competition with unlabeled promoter templates to characterize time dependent dissociation of the PIC in the presence and absence of RIP are problematic since the dissociation of the PIC is very fast at high temperatures, irrespective of RIP.

6. Does RIP bind to the RNAP bound to an elongation scaffold? Presumably not or poorly, if the claim that RIP competes with template DNA (page 18, line 4) is correct?

- We carried out additional binding experiments to elaborate on this point. Interestingly, the DNA/RNA nucleic acid template and RIP are able to bind simultaneously to the RNAP. Three lines of evidence support this finding: 1. RIP cannot compete for the template in preformed RNAP-template complexes, 2. The template cannot compete for RIP in preformed RNAP-RIP complexes, and most persuasively 3. The addition of RIP leads to a supershift of the RNAP-template complex forming a RNAP-RIP-template complex.
- We have added the following sentence to the manuscript on lines 340-347: *Alternatively, ORF145/RIP could inhibit elongation via an allosteric mechanism, e.g. by translating conformational changes via the clamp into the RNAP active site, or by preventing DNA movement along the DNA binding*

channel. EMSA supershift experiments using radiolabelled DNA/RNA elongation scaffolds and ORF145/RIP support an allosteric mechanism. When ORF145/RIP is added to RNAP-DNA/RNA complexes a new complex with lower mobility is formed in a dose-dependent fashion (Figure 5 D). We have introduced the following paragraph in the discussion section in lines 416-424: This binding site provides important clues to the molecular mechanisms of ORF145/RIP function. In the context of the PIC, ORF145/RIP could interfere with the binding of transcription factor TFB core domain or DNA within the DNA binding channel of RNAP, both would interfere with the PIC. EMSA experiments mimicking the TEC demonstrate that RIP can bind to RNAP in the context of the RNAP-DNA-RNA complex without disrupting it, while transcription elongation assays show that RIP efficiently inhibits elongation. This supports the notion that RIP binding to RNAP induces conformational changes that renders it inactive (Figure 7A).

7. The authors can experimentally demonstrate that RIP and DNA compete for clamp interaction by conducting an FA assay with labelled DNA and look for displacement of the DNA as a function of RIP concentration/time.

- RIP and DNA do not compete for RNAP binding (see above), and as discussed above, the required experimental conditions including high temperatures are not conducive to FA analysis of this complex.

8. As presented, I did find that the section containing the homology modelling data of RIP adds little to advance our understanding of RIP function. The authors propose that the acidic patch might be functionally important considering that it binds to the clamp (=DNA binding channel). The authors are urged to conduct a mutational analysis of this patch or of any other region of functional significance) to further validate their claims of how RIP binds to and inhibits the RNAP. Clearly, what is required is the structure of RIP - but this is well beyond the scope of the manuscript and it would be unfair to ask the authors for it.

- We follow the advice of reviewer 2 and have removed the isopotential surface representation of RIP and consequently the description of the negatively charged patch.
- We do think that the homology model contributes to the message of the manuscript in several ways, not only in terms of the structure-function relationship of RIP but in particular because the functional diversification of a major coat protein into a transcription factor is very interesting. We are in the process of crystallising RIP (no success yet!) and will in that paper include an extensive mutational analysis that we feel is beyond the scope of the current manuscript. However, we have now included one pivotal RIP mutant variant with a non RNAP-binding phenotype that if not proves the accuracy of the complete homology model at least supports it. The overall structure of RIP encompasses a compact core with one distinguishing feature - a C-terminal tail that protrudes from the core almost at a right angle. We have deleted this from the model predicted tail in the truncation mutant ORF145 Δ 114-145. This mutant variant is expressed at high levels and it is soluble and thermostable, which suggests that the truncation does not compromise the structural integrity of RIP - as can be predicted from the homology model. Binding and abortive transcription experiments reveal that this mutant cannot bind RNAP nor does it repress transcription. Altogether, this supports not only the predictive power of the homology model, but further shows that the protruding C-terminal motif is important for RNAP binding and RIP function.
- We have added the following paragraph to the results section of the manuscript on line 379-386: *In order to investigate the role of the C-terminal tail in RIP function we created a short C-terminal truncation variant (deletion*

highlighted in red in Figure 6 A). The ORF145/RIP^{D114-145} variant was expressed at high levels and proved to remain soluble after heat treatment, symptomatic of a correctly folded protein. This mutant was not able to compete with wild type RIP for the binding to RNAP (Figure 6 C) and likewise, unable to repress transcription (Figure 6 D), which suggests that the C-terminal tail of ORF145/RIP is required for RNAP binding and thus inhibition. And in the discussion section we have added line 493: In this context its important to note that the C-terminal deletion variant of RIP completely abrogated RNAP binding and inhibition (Figure 6 C and D).

9. If RIP or RNAP is purified from *Sulfolobus* (under conditions where RIP is expressed and exhibits toxicity) - does RIP co-elute with the RNAP (or vice versa) and is the co-eluted RNAP composition/activity different in the presence/absence of RIP?

- The transformation of a plasmid encoding the RIP open reading frame into *Sso* is lethal to the cell, even under conditions when RIP expression is not induced. This is described in the manuscript in lines 392-405.

Reviewer #2 (Remarks to the Author):

I enjoyed reading this interesting paper. Sheppard et al. describes the mechanism of action of a viral protein from the archaeal virus ATV (called RIP by the authors) in repressing the archaeal DNA directed RNA polymerase. The manuscript unfolds nicely.

The paper includes an experimental part using classical biochemistry, FRET and CD, and a theoretical part using homology modeling.

The experimental part argues that the RIP protein binds to the clamp occluding the DNA entry channel impeding the loading of the DNA template onto the RNAP. Thus it is a potent inhibitor of transcription. The modeling part makes the claim that this protein possesses a helix-bundle fold similar to the major capsid protein of ATV.

The work is original as this is the first example of a viral archaeal repressor acting directly on the RNAP however the mechanism of action proposed replicates similar mechanisms observed in bacteriophage. From the methodological point view the main techniques are largely used in the field.

Both the experimental and modeling parts left me with various questions and I would be interested in hearing the author's views on the points that I will raise below.

This work may be of some interest to the readership of Nature Communication, but that the authors have a chance to answer comments from me and other referees and that the authors be more specific about the interactions governing their proposed repression mechanism by RIP.

Specific comments:

Page. 4 - Line 50. The idea of three virospheres is misleading even within quotes. So far the virosphere remains one; please see the International Committee on Taxonomy of Viruses.

- *The concept of three independent virospheres is contested in the field, but we accept reviewer 2's comment and have modified the text in lines 48-50: Cellular organisms belonging to the eukaryotes, archaea and bacteria are accompanied by their cognate viruses (termed phages in bacteria) ⁴.*

Page 10. - Lines 173-176. Although others have previously generated rClamp from Pfu and visualize it in 3D by X-ray crystallography whether also Sso rClamp folds correctly is a plausible assumption. However, the Fig. 1D red trace (rClamp alone) doesn't show a symmetrical elution peak. In unfolding and re-folding of certain proteins the lack of a symmetrical peak with streaking tails is suggestive of protein not properly folded -

I wonder if this is the case here. Can the authors explain the reasons for these profiles? Also the peaks on the Figure 1D (elution profiles) do not seem centered respectively at 16 ml (red trace) and 14 ml (blue trace) as stated in the text but more likely to 14.2 ml (red trace) and 13 ml (blue trace) unless the authors used the gel below to define the 'peak' elution which would be a bit unusual. Also why not perform CD on the Sso rClamp to assess its secondary structure in solution? MALS analysis would also be beneficial here.

- *The recombinant clamp is expressed as soluble protein and is not reconstituted by denaturation-renaturation (unfolding-refolding). We agree with the assessment of reviewer 2 concerning the asymmetry of the SEC peak, and have noticed and described it in a previous publication where we attribute this aberrant behavior to weak nonspecific interactions with the SEC column matrix (Elife (2015); 4:e08378. doi: 10.7554/eLife.08378). However,*

considering that we provide positive evidence for its structural integrity (heat stability) and more importantly interaction properties – it binds the transcription factor TFE (eLife 2015) and RIP (current manuscript) we feel that CD and SEC-MALS would not contribute any additional information that would contribute to the message of the current manuscript. In particular because the interaction with the clamp is in agreement with 1. Far-western blotting, 2. XL-MS, and 3. smFRET experiments.

- The description of the elution peaks has been altered to define that of the chromatogram rather than the gel, lines 174-177.

Page 10. - Line 181. 'at higher resolution' appears a bit too grand - the authors have further mapped the potential binding sites of RIP onto the RNAP but not identified the interacting residues - I would suggest removing it.

Moreover the section ORF145/RIP binds in the DNA-binding channel describes more the protocol used for the experiment rather than the results of the experiment per se. For example there is no mention within the text of which lysines within RNAP and RIP have been found cross-linked. The re-direction to the figures helps partially as those on the RNAP have been displayed in red. Also the order of presentation of Figures within the text doesn't follow the A, B, C but the A, C, B panels which is illogical.

- We moderated the text by deleting 'at higher resolution' now in line 182
- A list of the RNAP-RIP cross-links has now been included in Figure 2C.
- The flow of the text has also been altered to follow A, B and C.

The authors on page 12 lines 209-210 explain the observed binding to Rpo8 as possibly due to the high protein concentrations. The question is: do the authors 'mix & match' RNAP and RIP for the cross-linking experiment or do they use the RNAP-RIP complex eluting from the SEC experiment? In the former case possible experiment duplicates at lower concentrations of RIP might untangle the described artifact; in the latter case no 'high protein concentration' issue should emerge.

- We combined highly purified RNAP and RIP prior to the addition of BS3 and crosslinking according to the published protocols. After again consulting with the Rappsilber lab the protein concentrations we used were not unusually high (as compared to many other targets they performed XL-MS on), and we removed this comment from the manuscript, now lines 237-239.

Fig 2C would benefit from being a larger size, in particular the network involving RIP with Rpo8 and Rpo1' with marking of the Lys residues involved. The remaining network to the opinion of this reviewer is a bit confusing and it can be displayed as Supplementary Figure. Finally there is no mention of the fact that RIP has 14 lysines and of these, only 5 cross-link and they seem to be centrally located in the sequence. Where would these lysines map onto the modeled RIP structure in Fig. 6A?

- On reviewer 2's suggestion we have limited the interaction network to that of ORF145/RIP and RNAP subunits and included the full version in the supplementary information in Figure S2A.
- Only two lysine residues in RIP, K58 and K74, form (multiple) cross-links with RNAP. We have provided an additional supplementary Figure S2B that maps all lysine residues on our homology model.

Page 13. - Line 236. Have the authors tested the possibility of RIP directly binding to DNA?

- Yes, we have directly tested the possibility of RIP binding to DNA using a range of nucleic acid templates in EMSA experiments and could not observe any interactions. In line with this observation its noteworthy that RIP cannot be purified using heparin columns.

Page 14. - Line 257. The majority of the assembled RNAP in solution appear to have the clamp in closed conformation as the distribution is 82% (close) vs 18% (open). Thus RIP would bind to both conformations and lead to an intermediate conformation - although in 80% of the cases this intermediated conformation would be reflecting the RIP binding to the closed RNAP - how much is the distance between the donor and acceptor in the intermediate conformation? And how does it compare in terms of distances with the close and open ones? Have I missed this information within the text?

- We have been hesitant to provide the exact distances in the manuscript since the distance measured between the dyes does not equal the absolute distance between the amino acids derivatised with the dyes. However, upon recommendation from the reviewer we now include the values listed below in the manuscript text.
 - RNAP wt open clamp (18%, low FRET population $E = 0.40$): 56 Å
 - RNAP wt closed clamp (82%, high FRET population $E = 0.67$): 46 Å
 - RNAP wt + RIP (intermediate conformation $E = 0.58$): 49 Å

- *line 258: In a previous study the immobilised RNAP shows two FRET populations with a major high FRET population at $E = 0.67 \pm 0.01$ (82±4%) corresponding to an inter-dye distance of 46 Å and a minor low FRET population at $E = 0.40 \pm 0.03$ (18±3%) corresponding to 56 Å. These can be assigned to a RNAP clamp in a closed and an open conformation, respectively (Figure 4B) ¹⁴. In contrast, the RNAP-ORF145/RIP complexes are radically different with only a single population of intermediate FRET efficiency ($E = 0.58 \pm 0.01$) (Figure 4B), corresponding to 49 Å.*

Also I was puzzled by the fact that Fig. 4B left appears to be the same as that below published by the same authors in reference 14 (see figure en attachment, Panel D below) except for some aesthetic differences. Am I right? There is nothing wrong in this as at page 14 lines 255-259 the reference 14 is indeed mentioned although the sentence would benefit from adding: "In a previous study the immobilized RNAP....., respectively (Figure 4B) 14." However, in the caption of Figure 4B - this is not mentioned - and it should be explicitly stated that Fig. 4B, left has been adapted from Reference 14.

- *The origin of the RNAP control has now been clarified (see above) and in the Figure 4B legend.*

Page 18 - Lines 335-338. These lines condense the essential findings of the manuscript, and are strongly supported experimentally.

Page 18 - Line 340 onwards. I am a bit confused in this section. The experimental data are the CD spectra whose deconvolution using algorithms in Dichroweb indicate that 65% adopt α -helical secondary structure. The homology model although suggestive remains a speculative model - a serious point is that there is no mention of how reliable the generated model is - do phyre2 and I-tasser have confidence indexes? Also I would not go so far as to calculate the isopotential surface from a model extrapolated on a sequence identity of less than 30% - BTW how was the isopotential surface calculated? - there is no mention in the methods.

- *The confidence indexes for the homology model have now been included in the main text lines 364-365: The I-TASSER confidence scores for this model (C-score -2.12, TM-Score 0.46 ± 0.15) are suggestive of a prediction with mid-range reliability as well as the Figure 6 legend.*

- We have followed the advice of reviewer 2 and removed the surface charge distribution in Figure 6.

Moreover at page 18 lines 345-348 - the "It seems unlikely that.....similar functions, but their....." appears redundant as in reference 33 it is already stated that products of ORF131 and ORF145 are paralogs.

- This text has been modified but we still feel it is necessary to state that ORF131 and ORF145 are paralogous for readers who are not familiar with reference 33.

This entire section seems an add-on on the biochemical and FRET analysis and only CD data are reliable - I read this section several times and could not figure out what reliable insight the modelling provides.

- Following the advice of reviewer 1 and 2, as mentioned above, we have expanded the section which provides structural insights into ORF145/RIP. The homology model revealed the presence of a short, flexible C-terminal tail. We have now included functional data on a C-terminal truncation mutant of this region that shows that this domain is crucial for RNAP binding and inhibitory activity.
- *Line 379: In order to investigate the role of the C-terminal tail in RIP function we created a short C-terminal truncation variant (deletion highlighted in red in Figure 6 A). The ORF145^{delta114-145} variant was expressed at high levels and proved to remain soluble after heat treatment, symptomatic of a correctly folded protein. This mutant was not able to compete for the binding of wild type RIP to RNAP (Figure 6 C) and likewise, unable to repress transcription (Figure 6 D), which suggests that the C-terminal tail of RIP is required for RNAP binding and thus inhibition.*

Page 21 - Lines 388 onwards. The discussion is clearly laid out but there are some lapses in assumptions.

At line 401 - 'RIP could act like....' would be better 'RIP might act like...' as the evidence for the mechanism in vivo have not been ascertained.

- *Line 429: Sentence changed to 'might' instead of 'could'.*

The evidence on how it would work comes from in vitro assembled RNAP whose 80% possess a closed clamp and 20% open. Difficult to extrapolate this observation to the transcription cycle. Also very little information is given on how this intermediate conformation would look like from smFRET data. Does 'intermediate' means between closed and open clamp? Again how much in ångström?

- The smFRET essay measures the change of distance between two dyes incorporated into two RNAP subunits across the DNA binding channel. It provides persuasive evidence that this distance changes upon RIP binding, but cannot describe the precise structural changes that RIP induces in RNAP, which could involve simple hinge-like movements of the clamp or more extensive rearrangements between the two largest RNAP subunits e.g. akin to the pivoting/ratcheting described in the bacterial RNAP-Gfh1 complex
- See discussion above regarding the smFRET and distance measurements, and for accurate values line 258 in the manuscript.

At lines 406-408: the experimental data on the RIP structure are the CD spectra, the rest is speculation, including the isopotential surface for the reasons above expressed.

- See above, the isopotential surface has been removed from Figure 6

The rest of the Discussion flows amenable and is based on the biochemical data gathered which appear indeed more solid and informative than the modeling and also somehow of the smFRET outcome. Moreover the sequence homology searches and the phylogentic tree of major capsid proteins and RIP-like proteins looks interesting but it proves that despite the sequence homology it is very hard to get insight into function and I would say structure as well.

What would have strengthened the manuscript is the identification of those residues onto RIP and/or RNAP that indeed interact. Do the authors know if is the N- or C-terminal tail of RIP that interacts with RNAP or the central body? Have they tried to assess by RIP truncation which portion of the protein is the interacting one? The cross-linking experiments although very useful do not directly provide this information.

The mechanism shown in Figure 7A is appealing for its simplicity but at the same time doesn't say much about specific interactions which would greatly help to understand at molecular level how the inhibition occur in the light also of the others transcription binding factors.

- We have changed Figure 7 A to include a better overview of RIP action in the (i) free RNAP, (ii) PIC, and (iii) elongation complex, even though the precise details of the structural mechanism remain uncharted until the RNAP-RIP structures have been solved.

Thanks again for the chance to read this manuscript. This is undoubtedly well-presented and it reads amenable but there is a decided imbalance between the experimental results and how far these results have been interpreted.

Minor points:

-) Reference 14 was published in March 2015 - it should be updated.
-) Page 26 - line 504. A full stop should precede 'Sso TFE.....'
-) Page 33 - line 633: is it said why the '65 oC and 20 min' incubation conditions were used?

- All minor points have been rectified in the text.

Reviewer #3 (Remarks to the Author):

The authors describe a protein of Acidianus two-tailed virus (ATV) that interacts with host RNA polymerase to inhibit transcription. They characterize this interaction using a variety of different strategies, all supporting the conclusion that the ORF145 protein binds to the DNA binding channel of RNAP, locking the flexible RNAP clamp into a fixed position. Once bound to RNAP, transcription of both host and virus promoters is blocked.

The study is significant in that it is the first description of a virus-encoded transcription factor that acts directly on host RNAP, rather than blocking its access to promoter sites. The manuscript was a pleasure to read, and I find the results of this multidisciplinary approach persuasive, and the data presented in a clear and concise manner. I believe the conclusions are valid, and I have only a few comments or suggestions that I think will improve the manuscript.

1. Lines 86/92. It would be useful to reference at least one of the early studies of gene regulation in archaeal viruses. I think the Ken and Hackett study of phiH repressor (Ken, R. & N.R. Hackett, (1991) Halobacterium halobium strains lysogenic for phage phi H contain a protein resembling coliphage repressors. J Bacteriol 173: 955-960.) is probably the earliest.

➤ This reference has now been included.

2. I was initially concerned that the 32P labelling of RIP might alter its binding properties, but the results obtained with this label fit in so well with data from unlabelled or fluor labeled RIP that I am persuaded that this was not an issue.

➤ We did experimentally test whether the 32P labeling would affect the activity of RIP, and it did not abolish nor weaken the inhibitory activity in e.g. promoter directed transcription assays.

3. Line 291: I think 'indicated' is too strong a term given the lack of direct experimental proof at this stage. I would prefer 'suggested' or 'consistent with' or similar terminology.

➤ We have altered the text as suggested.

4. Line 392: I don't think XL is defined before this, and assume you mean cross-linking. Either define this abbreviation when you first use the term or, better, just use cross-linking MS to make it clear.

➤ We have altered the text as suggested.

5. In the discussion regarding levels of RIP and its influence on transcription, is there anything known about the transcription pattern of ORF145 or the half-life of RIP? For example, if it is only expressed at a specific time during infection, or if the protein degrades rapidly in cells. If there is published data on these aspects then they would be useful here.

➤ Unfortunately, the expression profile of RIP (e.g. as a function of time post infection) and its half life is unknown.

6. Line 468: this describes an experimental study, and should not be part of the Discussion section. If the authors do not wish to integrate this into the results section, then I think it should be removed.

➤ This section of the discussion refers briefly to our findings concerning the ORF114 and ORF122 proteins because it is relevant to our phylogenetic analysis and emphasizes our attempts to pursue in vivo studies of RIP-like proteins. We do not actually present any results here (for reasons of clarity)

and therefore we placed it into the discussion section. However, if the editor and reviewer find it strictly required to relocate this paragraph we will obey.

7. Line 672: I find this sentence difficult to follow, and it would be clearer if the component concentrations of the synthetic scaffold were stated more clearly. For example, the abbreviations TS-DNA and NTS-DNA do not seem to be given in full in the manuscript.

➤ This text has now been modified according to reviewer 3's recommendation.

8. References: some attention needs to be taken with the references. Species names in italics (e.g. ref 16), and some variation in formatting (e.g. journal names in sentence or title case, such as refs 5 and 6).

➤ The references have now been altered.

9. Figure 1, panel C: I think it would be clearer to the reader if a bit more description of the Far-Western blot was given, including the antibody used for detection.

➤ We have included further details in the figure legend. A full description of the methodology is available in the methods section.

10. Figure 3, panel A: It would be clearer to me if the complementary bases of the two strands were indicated with vertical bars or dots between them. My initial impression looking at this sequence was that it was a single nucleotide sequence split over two lines. Indicating 5' and 3' ends might also be useful.

➤ This part of the figure has now been modified.

11: Line 1027: I think it is Phyre2 rather than a superscript 2.

➤ This has been corrected.

REVIEWERS' COMMENTS:

Reviewer #1 (Remarks to the Author):

The authors have done a fine job with the revision and have addressed the points raised by all reviewers comprehensively (where technically and logically possible). I have no further comments to make and the manuscript can now be accepted for publication in NC. However, I strongly recommend that the authors include the word 'archae' or 'archaeo-viral' in the title.

I wish to make the following suggestions, which the authors might want to consider moving forwards as future work (please note that the following are merely suggestions and are not a pre-requisite for acceptance of this MS):

1. Since the plasmid encoding RIP is toxic then the authors might want to consider doing a 'pull-down' assay with *Sso* cell extract and immobilised RIP. The authors should also consider purifying the RNAP from infected cells as a function of time and investigate if and when RIP is present and if RNAP undergoes any PTM. This might provide exciting new insights into archaeal viruses.
2. Since RIP can bind to the RNAP-elongation scaffold presumably its MOA can be multi pronged/faceted? This is worth investigating in detail.

Reviewer #3 (Remarks to the Author):

I have read all the reviewer comments and the authors responses to these, and have re-read the entire manuscript. I find the authors have adequately addressed all the comments that have been raised, and have made appropriate revisions to the manuscript.

Two minor comments:

1. I wonder if the superscripted reference (44) on line 566, at the end of the equation, would be better placed earlier in the sentence. Then there would be no possible confusion that the 44 is an exponent of the equation rather than a citation.
2. The references still have many species names not in italics. Please make sure this is correct in the final version.

Point-by-point response to reviewers comments

Reviewer #1 (Remarks to the Author):

The authors have done a fine job with the revision and have addressed the points raised by all reviewers comprehensively (where technically and logically possible). I have no further comments to make and the manuscript can now be accepted for publication in NC. However, I strongly recommend that the authors include the word 'archae' or 'archaeo-viral' in the title. I wish to make the following suggestions, which the authors might want to consider moving forwards as future work (please note that the following are merely suggestions and are not a pre-requisite for acceptance of this MS):

- Under the recommendation of the editor and reviewers the title has now been updated to include the name ORF145/RIP rather than simply RIP but we have chosen to exclude the term 'archaeo-viral' in order to appeal to a wider audience.
 1. Since the plasmid encoding RIP is toxic then the authors might want to consider doing a 'pull-down' assay with Sso cell extract and immobilised RIP. The authors should also consider purifying the RNAP from infected cells as a function of time and investigate if and when RIP is present and if RNAP undergoes any PTM. This might provide exciting new insights into archaeal viruses.
 2. Since RIP can bind to the RNAP-elongation scaffold presumably its MOA can be multi pronged/faceted? This is worth investigating in detail.
- We thank the reviewer for their suggestions for future work. Some of these lines of investigation are already underway in the lab.

Reviewer #3 (Remarks to the Author):

I have read all the reviewer comments and the authors responses to these, and have re-read the entire manuscript. I find the authors have adequately addressed all the comments that have been raised, and have made appropriate revisions to the manuscript. Two minor comments:□

1. I wonder if the superscripted reference (44) on line 566, at the end of the equation, would be better placed earlier in the sentence. Then there would be no possible confusion that the 44 is an exponent of the equation rather than a citation.
- The reference has now been moved to an earlier point in the sentence to avoid confusion.
 2. The references still have many species names not in italics. Please make sure this is correct in the final version.
 - Species names are now in italics.